# Efficient Retrieval-Augmented Generation with Deferred Positional Encoding

## Abstract

Key-value (KV) caching accelerates inference in large language models (LLMs) by reusing computations from previously generated tokens. Its importance becomes even greater in long-context applications such as retrieval-augmented generation (RAG) and in-context learning (ICL). However, conventional KV caching embeds positional information directly into the cache, limiting its reusability. Existing solutions face a memory-compute trade-off. Specifically, they either restrict reuse to prefixes or require expensive memory materialization for position adjustment. We introduce Lazy-Attention, a novel attention mechanism that kernelizes deferred positional encoding to enable zero-copy, position-agnostic KV reuse. By fusing positional adjustment into the attention kernel on-the-fly, Lazy-Attention resolves the materialization bottleneck, allowing a single physical KV copy to serve multiple logical requests at arbitrary positions. Leveraging two optimized kernels tailored for prefilling and decoding, Lazy-Attention achieves significant efficiency improvements: under skewed document distributions, it reduces time-to-first-token (TTFT) by $1.37\times$ and increases inference throughput by $1.40\times$ compared to the state-of-the-art Block Attention, while maintaining comparable output quality.

## 1 Introduction

Retrieval-augmented generation (RAG) (Lan, 2024; Guu et al., 2020; Lewis et al., 2020) improves the accuracy and timeliness of responses (Ram et al., 2023; Asai et al., 2023) by enriching user queries with external data (Xiong et al., 2024; Gao et al., 2024; Kalra et al., 2024). However, processing this external data remains a major bottleneck for achieving low-latency RAG, since answer generation can only begin once the data has been fully processed. This step scales poorly—the computational complexity grows quadratically with input length (Jiang et al., 2024). The overhead will further worsen as modern models support increasingly longer context windows (Team et al., 2024). While each retrieved document must be processed at least once, its results could be stored in a form that enables more flexible reuse. This motivates a careful investigation into *reusable components*.

Existing approaches use conventional key-value (KV) cache (Qin et al., 2024) as reusable components, but remain ineffective due to their *position-awareness*. That is, cached values are reusable only if the associated data (e.g., a document) appears in the same position as before. This restriction is considered by earlier caching techniques, which accordingly focus on the identification of exactly matching document sequences (Kwon et al., 2023a; Gim et al., 2024; Jin et al., 2024). However, the chance of observing an exact match is lower than that of encountering individual documents. Recent works (Lu et al., 2025; Ma et al., 2025) show that KV cache can be reused even for the documents appearing in different positions if their positional information is re-encoded. This enhances reusability but is still memory-inefficient because position re-encoding requires duplicating the KV cache. In-place updates can cause race conditions and incorrect outcomes if two prompts in the same batch share the same data. This limitation—position-awareness—makes caching ineffective. To achieve 100% cache hit for $N$ documents, existing methods need $O(N!)$ space, which is impractical. Caching can be more effective if a cached item can be reused across multiple prompts.

In this paper, we argue that reusable components should be **position-agnostic** to enable effective caching under limited memory. On modern GPUs, High Bandwidth Memory (HBM) remains insufficient for storing large volumes of KV cache. While more space-efficient KV representations have

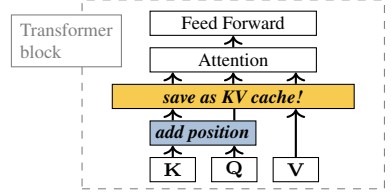
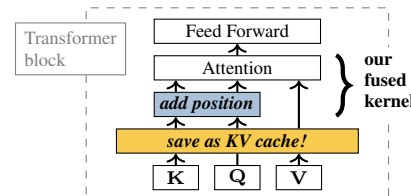

(a) Existing positional encoding. Positional encoding (blue box) is *eagerly* applied before saving (yellow box) the computed key (K) and queries (Q).

(b) Our proposed positional encoding. Positional encoding (blue box) is *lazily* applied inside our fused attention kernel. KV cache omits positions.

Figure 1: Comparison of positional encoding strategies. Existing methods, even the state-of-the-art Block-Attention for KV cache reuse, eagerly apply positions before caching, making KV caches position-dependent and non-shareable. Lazy-Attention applies positions lazily inside the fused attention kernel, enabling *reuse and sharing* without duplication.

been proposed—ranging from compression (Ge et al., 2024; Liu et al., 2024) to grouping (Ainslie et al., 2023) and latent representation (DeepSeek-AI et al., 2024)—HBM is fundamentally limited in holding exponentially many KV caches. If there is a mechanism that can allow KV cache reuse independent of document positions, it could dramatically improve cache effectiveness. Specifically, achieving 100% cache hit for $N$ documents would require only $O(N)$ space, which is $O((N-1)!)$ times more space-efficient than existing methods. This would also make it feasible to preprocess and fully cache a medium-sized database using a combination of GPU HBM and host memory, enabling near-zero latency for a wide range of RAG applications. However, this possibility is nearly impossible with existing mechanisms. Positions are *eagerly* embedded into KV cache for nearly all models (e.g., BERT (Devlin et al., 2019), Llama (Dubey et al., 2024), DeepSeek (DeepSeek-AI et al., 2024)) running on the latest serving systems (e.g., vLLM (Kwon et al., 2023b), Orca (Yu et al., 2022), SGLang (Zheng et al., 2024)). Achieving *position-agnostic* reuses will therefore require a fundamentally different approach.

We propose Lazy-Attention, an attention mechanism that can enable **position-agnostic** KV reuse, significantly improving cache effectiveness for RAG applications. While the high-level concept of decoupling RoPE from KV storage has been explored (Lu et al., 2025; Ma et al., 2025), prior attempts faced a critical Memory–Compute trade-off: they either required materializing position-adjusted copies (high memory/bandwidth cost) or restricted reuse to prefixes (low flexibility). Lazy-Attention resolves this trade-off by **kernelizing** the RoPE-decoupling technique. The key idea shown in Fig. 1 is to develop an alternative form of KV cache that can be shared among multiple prompts *without* approximating attention computation. Lazy-Attention achieves this by *deferring*, rather than omitting, positional encoding until the final stage—when the attention is computed. At that point, the kernel dynamically encodes positions on chip by considering relative distances between each query-key pair, requiring only a few additional kernel variables. Importantly, positional encoding in Lazy-Attention is only **transient**, lasting only for the brief period of attention computation. This design completely avoids additional data materialization in HBM. Despite the difference, Lazy-Attention's attention is mathematically identical to existing methods that duplicate KV caches for position re-encoding (Lu et al., 2025; Ma et al., 2025). As a result, it produces the same attention scores and generated answers.

We implement Lazy-Attention within vLLM (Kwon et al., 2023b) by developing two tailor-made Triton kernels (Tillet et al., 2019) for prefilling and decoding. Designing separate kernels is non-trivial, as prefilling is typically computation-bound while decoding is memory-bandwidth-bound. To address these distinct bottlenecks, our kernels are carefully optimized to minimize extra computation in prefilling and extra I/O in decoding. Furthermore, our custom attention mechanism integrates positional encoding directly into the key-value multiplications. Despite this added logic, the runtime overhead remains around 0.2% in practice, while substantially improving cache effectiveness and reducing latency.

We empirically validated Lazy-Attention on diverse benchmarks, including CacheBlend (Yao et al., 2025), Prompt Cache (Gim et al., 2024), and Block-Attention (Ma et al., 2025), demonstrating consistent improvements in latency and throughput. Our approach is particularly effective in common RAG scenarios with skewed document access patterns, where hot documents are frequently

reused. Compared to the state-of-the-art Block Attention, Lazy-Attention reduces time-to-first-token (TTFT) by up to 1.37× and increases inference throughput by up to 1.40×, all while maintaining similar output quality. Our contribution can be summarized as follows.

- We propose a novel attention mechanism that kernelizes the RoPE-decoupling technique to enable zero-copy, position-agnostic KV reuse. By deferring positional encoding to a transient step within the attention kernel, our method resolves the memory–compute trade-off that previously hindered the scaling of arbitrary-position reuse.

- We demonstrate that this position-agnostic design significantly increases the cache hit ratio. A single cached document entry can be shared across all requests, regardless of its position, maximizing cache efficiency under memory constraints. In scenarios with skewed access patterns, the hit ratio improves by 7.2× compared to naive prefix caching.

- We provide highly optimized Triton kernels for both prefill and decoding that implement our mechanism with negligible overhead (around 0.2%) even for long-context inputs, translating our architectural improvements into substantial practical gains in end-to-end throughput and latency for RAG workloads.

## 2 BACKGROUND AND MOTIVATION

In this section, we briefly review RAG and KV cache reuse, then discuss why existing techniques struggle with efficiency under dynamic contexts.

Retrieval-augmented generation (RAG) (Lan, 2024; Guu et al., 2020; Lewis et al., 2020; Chan et al., 2025) augments LLMs with external knowledge. Given a query $\mathcal{Q}$, a retriever selects top-$N$ relevant documents $\mathcal{D} = \{d_1, \ldots, d_N\}$, which the generator conditions on when producing an answer. The most common approach is simple concatenation of retrieved documents (Izacard & Grave, 2021), though more complex schemes exist (Borgeaud et al., 2022).

RAG enables flexible knowledge grounding but also introduces variability: irrelevant or low-quality documents may need to be inserted or removed across queries (Li et al., 2023; Yoran et al., 2024; Cuconasu et al., 2024). Such changes alter the prefix, forcing standard attention to re-run the costly prefill step. This motivates techniques for reusing previously computed KV caches to support dynamic contexts efficiently.

It is desirable to cache documents independently and reuse them across queries. Existing methods—Prompt Cache (Gim et al., 2024), CacheBlend (Yao et al., 2025), and Block-Attention (Ma et al., 2025)—all rely on re-encoding positional information. While effective to some extent, this coupling between content and position leads to inefficiency: each distinct ordering of documents requires a separate copy of their KV caches. In the worst case, achieving 100% cache hit for $N$ documents demands $O(N!)$ space, far exceeding GPU memory capacity.

Relative positional encodings such as RoPE (Su et al., 2024) suggest a possible remedy: keys can be rotated on the fly to align with new positions. However, applying rotations dynamically during decoding requires recomputing large key matrices for every token, leading to prohibitive overhead.

In short, current approaches either exhaust memory with redundant cache copies or incur heavy computation from repeated position adjustments. This raises the central question: *can we achieve high cache hit ratios under limited memory without introducing large runtime overhead?*

## 3 LAZYATTENTION: ALGORITHM AND ANALYSIS

In this section, we show how the positional encoding for multiple documents can be computed *during* the attention calculation and seamlessly integrated with FlashAttention (Dao et al., 2022; Dao, 2024) tiling to achieve efficient computation.

**Problem Statement**  Motivated by the limitations of existing approaches and the constraints of GPU memory, the problem is to develop a method that efficiently manages and manipulates the positional information of document KV caches *on the fly* during attention computation, without incurring high recomputation or extensive copying costs. The method should correctly handle the

global positional alignment required for the query's attention and support scenarios involving multiple queries accessing the same document at different global offsets within a batch.

### 3.1 DEFERRED POSITIONAL ENCODING

In standard Transformer architectures, positional encoding is applied before attention computation, forcing explicit duplication of the KV cache whenever document positions differ. Our approach *defers* positional encoding until the attention step, ensuring the KV cache remains position-agnostic.

**Standard Attention**    In the standard formulation, the query, key, and value matrices, $\mathbf{Q}$, $\mathbf{K}$, and $\mathbf{V}$, are computed from the input sequence:

$$\text{Attention}(\mathbf{Q}, \mathbf{K}, \mathbf{V}) = \text{softmax}\left(\frac{\mathbf{Q}\mathbf{K}^T}{\sqrt{d_k}}\right)\mathbf{V}, \tag{1}$$

where $d_k$ is the dimensionality of $\mathbf{K}$.

Positional encoding allows the model to exploit sequence order when computing attention scores. A variety of methods exist—ranging from absolute to relative encodings such as XLNet, T5, RoPE, and iRoPE. Among these, **RoPE** (Rotary Positional Embedding) has become the most widely adopted due to its simplicity and effectiveness. In this paper, we focus on RoPE, though our method can be easily extended to other relative encodings. With RoPE, Equation 1 becomes:

$$\text{Attention}(\mathbf{Q}, \mathbf{K}, \mathbf{V}) = \text{softmax}\left(\frac{f(\mathbf{Q}, \mathbf{m})f(\mathbf{K}, \mathbf{n})^T}{\sqrt{d_k}}\right)\mathbf{V}, \tag{2}$$

where $f$ denotes the positional embedding function, and $\mathbf{m}$, $\mathbf{n}$ represent the positional indices of the tokens in $\mathbf{Q}$ and $\mathbf{K}$.

**Deferred Encoding**    In our method, the query, key, and value matrices are first computed from the input sequence without positional information. Positional encoding is applied only *during* attention computation, after retrieving entries from the KV cache. This means the cache stores purely content-based keys and values, which can be reused across arbitrary positions. The positional adjustments are dynamically applied at runtime, yielding both mathematical correctness and significantly improved cache efficiency.

**Example 1.** *Suppose a query $q$ encodes its positional index based on document length. Consider two documents, $d_1$ and $d_2$, with independently generated KV caches $c_1$ and $c_2$, each starting from position $0$. When reusing $c_2$ for $q$ after processing $d_1$, we rotate $q$ backwards by $|d_1|$ before computing attention, ensuring positional consistency.*

**Naïve deferred prefilling/encoding is expensive**    A straightforward implementation would follow the idea of *BlockAttention* show in Fig. 2(a): whenever we load a KV block, we first rotate $\mathbf{K}$ back to position $0$ and then rotate it forward to the target position.[1] Let $M$ be the query tile height (BLOCK_M), $N$ the KV tile width (BLOCK_SIZE), and $D$ the head size. The baseline attention per block performs *QK* and *PV* GEMMs with $4MND$ FLOPs (counting mul+add as two FLOPs). Each rotary application costs 6 FLOPs per element (four muls + two adds), so *two* rotations of $\mathbf{K} \in \mathbb{R}^{D \times N}$ introduce

$$\Delta\text{FLOPs}_{\text{naïve}} = 2 \times 6\,DN = 12\,DN, \qquad \frac{\Delta\text{FLOPs}_{\text{naïve}}}{\text{baseline}} = \frac{12DN}{4MND} = \frac{3}{M}.$$

Thus, the relative overhead is $\frac{3}{M}$: for **decoding** where $M = 1$, this is a $+300\%$ FLOPs increase; for **prefill** with $M = 128$, the extra FLOPs are still $+2.34\%$. On the I/O side, each rotary needs a $\cos/\sin$ vector of length $D$, hence two rotations add $2D$ elements per block. Relative to reading $\mathbf{K}$ and $\mathbf{V}$ (about $2DN$ elements per block), the extra bandwidth fraction is

$$\frac{\Delta\text{I/O}_{\text{naïve}}}{\text{baseline I/O}} \approx \frac{2D}{2DN} = \frac{1}{N}.$$

With a typical $N = 64$, this is $+1.56\%$ bandwidth per block, sustained across *all* blocks.

---

[1] Equivalently, two successive rotary transforms per block.

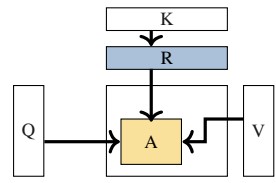 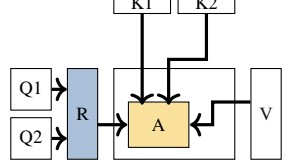 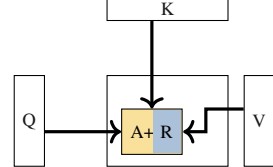

(a) Existing: keys and queries are rotated (R) before attention (A)

(b) Our prefill: only queries are rotated before attention

(c) Our decoding: rotation occurs inside the fused attention kernel

Figure 2: Where to apply rotary (R) in tiled attention. (a) Existing kernels rotate keys twice before attention (**A**), tying the KV cache to absolute positions. (b) Lazy-Attention (prefill): only the queries are rotated before **A** while keys/values are read as-is; single-sided rotation avoids KV duplication ($Q_1/Q_2$ denote the two half-dims used by RoPE, likewise $K_1/K_2$). (c) Lazy-Attention (decoding): rotation is deferred and fused inside the attention kernel, applied on-the-fly per tile using the relative offset, keeping the KV cache position-agnostic with negligible overhead.

**Why this gets amplified on GPUs** Modern attention kernels are heavily tuned and often operate near a hardware bound (compute- or bandwidth-limited). Any additional work inserted into the *inner* loop—especially extra global loads with poor locality—can push the kernel deeper into the bound and *amplify* the end-to-end cost. In practice, even a *single* extra scalar load inside the decoding inner loop (with a large address stride or low L1/L2 reuse) can degrade latency by $> 10\%$ due to lost coalescing, pipeline stalls, and reduced effective warp throughput. By contrast, our design avoids the double rotation of **K** entirely: we either (i) rotate at most *once* using the relative offset, or (ii) rotate the much smaller object (the query, when beneficial), and (iii) trigger rotation only on the small fraction of blocks that actually need it. This keeps both FLOPs and I/O overheads negligible and preserves the roofline position of the kernel.

If one insists on rotating **K**, using a *single* relative rotation instead of "back-to-zero then forward" halves the rotary cost to $6DN$, i.e., a relative overhead of $\frac{3}{2M}$ (still $+150\%$ for decoding with $M = 1$). Therefore, it is necessary for us to find an efficient and hardware-aware method to implement deferred positional encoding with low overhead.

## 3.2 Rotation within Tiling

We implement Lazy-Attention inside a tiled attention kernel with negligible overhead. Let $M$ and $N$ denote the query and KV tile sizes (rows and columns), and let $D$ be the head dimension. We use rotary positional encoding (RoPE), which only depends on relative positions.

**Fact 1** (Relative rotation for RoPE)**.** *RoPE can be represented as a rotation matrix* **R** *applied to queries and keys, with the rotation angle determined by token positions. For any token pair from* **Q** *and* **K** *with positions* **m** *and* **n**,

$$(\mathbf{R_m q})^\top (\mathbf{R_n k}) = \mathbf{q}^\top \mathbf{R_m^\top R_n\, k} = \mathbf{q}^\top \mathbf{R_{n-m}\, k},$$

*i.e., attention depends only on the* relative *offset* $\mathbf{n} - \mathbf{m}$.

**Design choice: move $Q$ or move $K$** To change the relative position between $(Q, K)$ we may rotate either $Q$ or $K$. Rotating both is unnecessary and increases cost; thus we anchor one and adjust the other. The efficient choice depends on the kernel regime:

**Prefill (compute-bound)** Prefill tiles are typically tall ($M \gg N$), hence *rotating $K$ is cheaper than rotating $Q$*. We apply a *single* relative rotation with offset $\Delta$ (never "back-to-zero then forward") like Fig. 2(b), combining the two half dimensions as

$$k_1' = k_1 \cos\Delta - k_2 \sin\Delta, \qquad k_2' = k_1 \sin\Delta + k_2 \cos\Delta.$$

This introduces 6 FLOPs per element of $K$ (four muls + two adds), i.e. $\Delta$FLOPs $= 6DN$ per KV tile. Relative to the baseline per-tile cost $4MND$ (QK and PV), the prefilling overhead is

$$\text{Overhead}_{\text{prefill}} \;=\; \tfrac{6DN}{4MND} \;=\; \tfrac{3}{2M}$$

(e.g., $M{=}128 \Rightarrow 0.59\%$). We table-drive $\cos/\sin$ and load a $D$-length row once per tile, adding only $\approx D/(2DN) = 1/(2N)$ bandwidth (e.g., $N{=}64 \Rightarrow 0.78\%$), and avoid SFU calls.

**Decoding (bandwidth-bound)** Decoding tiles are short ($M \approx 1$), so rotating $K$ would cost $6DN$ per tile; instead we *rotate Q only when needed*. Let $r$ be the fraction of KV tiles whose relative offset $\Delta \neq 0$ (a uniform per-tile condition likel Fig. 2(c), thus no warp divergence). A single $Q$ rotation costs $6MD$, giving per-trigger overhead $(6MD)/(4MND) = \frac{3}{2N}$ (e.g., $N{=}64 \Rightarrow 2.34\%$). Averaged across all tiles, this becomes Overhead$_{\text{decode}} = r \cdot \frac{3}{2N}$, which is negligible for small $r$ (e.g., $r{=}1\% \Rightarrow 0.023\%$). To eliminate extra memory traffic inside the inner loop, we pack (`block_id`, `offset`, `mask`) into a single 64-bit entry and recover them with register shifts, avoiding additional global loads. When position sets are large, we either cache $\cos/\sin(\Delta)$ across adjacent tiles with identical $\Delta$ or table-drive them; on-the-fly SFU is avoided in the hot path.

**What we *do not* do** The naïve scheme that rotates $K$ twice per tile (back to zero, then to the target) adds $12DN$ FLOPs per tile and $\approx 2D$ extra elements of $\cos/\sin$ loads, i.e., relative overheads $\frac{3}{M}$ in FLOPs and $\frac{1}{N}$ in bandwidth (e.g., $+300\%$ FLOPs for decoding with $M{=}1$). Our design always uses a *single* relative rotation.

### 3.3 ANALYSIS: COMPLEXITY OF LAZYATTENTION

**Computation** Lazy-Attention preserves the asymptotic complexity of standard attention. For sequence length $L$ and head size $D$, a single head performs $\mathcal{O}(L^2D)$ work for QK/PV; linear projections add $\mathcal{O}(Ld_{\text{model}}D)$ but are unchanged by our method. Deferred rotation contributes only constant-factor overheads described above: $3/(4M)$ for prefill and $r \cdot 3/(2N)$ for decoding.

**Memory** We store a *position-agnostic* KV cache, so keys/values are cached once per token: $\mathcal{O}(LD)$ per head. Approaches that embed absolute position into cache content and re-materialize under shifts can inflate memory by a factor proportional to the number of reused offsets; decoupling position from KV eliminates such duplication and keeps the footprint linear in $L$.

### 3.4 ANALYSIS: TTFT AND DECODING OVERHEAD

We decompose the time-to-first-token (TTFT) into two stages: (i) the *prefill* pass over the prompt and (ii) the *first decoding* step. Let $L$ be the prompt length, $D$ the head size, and $(M, N)$ the query/KV tile sizes. We adopt a simple roofline model: for a kernel with FLOPs $F$ and bytes $B$, its time is $\max(F/\mathcal{P}, B/\mathcal{B})$, where $\mathcal{P}$ and $\mathcal{B}$ denote sustained compute throughput and memory bandwidth (including efficiency factors).

**Prefill.** A tiled attention block performs $4MND$ FLOPs (QK+PV). With deferred rotation, we rotate *once* per KV tile (relative offset), incurring $6DN$ extra FLOPs for the $K$ combination (four muls + two adds per element). Thus the per-tile FLOPs overhead is $\frac{6DN}{4MND} = \frac{3}{2M}$, e.g., $M{=}128 \Rightarrow 1.18\%$. We table-drive $\cos/\sin$ and load a single $D$-length row per KV tile, adding $\approx D$ elements to the tile I/O. Relative to reading $K$ and $V$ ($\approx 2DN$ elements), the bandwidth overhead is $\frac{1}{2N}$ (e.g., $N{=}64 \Rightarrow 0.78\%$). Aggregating over tiles preserves these ratios; hence the prefill time is

$$T_{\text{prefill}} \approx \max\left(\tfrac{F_{\text{base}}}{\mathcal{P}}, \tfrac{B_{\text{base}}}{\mathcal{B}}\right) \cdot \left(1 + \tfrac{3}{2M}\right) \quad \text{with} \quad F_{\text{base}} \approx 4L^2D.$$

**First decoding step** Decoding is typically bandwidth-bound. For a single new query ($M{=}1$), the baseline work over $L$ keys/values is $F_{\text{base}} \approx 4LD$ FLOPs. We rotate the *query* only when the per-tile relative offset is nonzero. Let $r \in [0, 1]$ be the fraction of KV tiles with nonzero offset (a uniform per-tile condition, so no warp divergence). A single query rotation costs $6MD{=}6D$ FLOPs for that tile, i.e., a per-tile overhead of $6D/(4ND) = \frac{3}{2N}$. Averaged across all tiles, decode overhead $= r \cdot \frac{3}{2N}$ (e.g., $r{=}1\%$, $N{=}64 \Rightarrow 0.023\%$). Because offset and mask are packed into one 64-bit entry per KV tile and unpacked with register shifts, deferred rotation introduces *no* extra global loads inside the inner loop; query rotation is performed in registers. Hence $B_{\text{decode}}$ remains effectively unchanged:

$$T_{\text{decode(1st)}} \approx \max\left(\tfrac{4LD}{\mathcal{P}}, \tfrac{B_{\text{base}}}{\mathcal{B}}\right) \cdot \left(1 + r \cdot \tfrac{3}{2N}\right).$$

**TTFT bound and comparison** Combining both stages,

$$\text{TTFT} \approx T_{\text{prefill}} + T_{\text{decode(1st)}} = \max\left(\frac{4L^2D}{\mathcal{P}}, \frac{B_{\text{prefill}}}{\mathcal{B}}\right)\left(1 + \frac{3}{2M}\right) + \max\left(\frac{4LD}{\mathcal{P}}, \frac{B_{\text{decode}}}{\mathcal{B}}\right)\left(1 + r \cdot \frac{3}{2N}\right).$$

In contrast, a naïve scheme that rotates $K$ *twice* per KV tile (back-to-zero and forward-to-target) adds $12DN$ FLOPs per tile, i.e., $\frac{3}{M}$ overhead in prefill (e.g., $M{=}128 \Rightarrow 2.34\%$) and $\frac{3}{M}$ in decoding (catastrophic when $M{=}1$). Our method avoids the double rotation, keeps bandwidth unchanged in decoding, and ensures all rotation branches are uniform at tile granularity (no warp divergence), leading to TTFT close to the baseline roofline.

## 4 EVALUATION

In this section, we evaluate Lazy-Attention along four research questions (RQs) to demonstrate its advantages and analyze its overhead. More experimental details can be found in Section B.

**RQ1** Does Lazy-Attention reduce time for the first token (TTFT) under different serving loads?

**RQ2** For repeated documents across requests, does Lazy-Attention attain a high KV hit ratio with limited GPU memory for KV cache?

**RQ3** What is the latency overhead of the extra deferred rotation operation of Lazy-Attention in prefill and decoding respectively?

**RQ4** Does Lazy-Attention preserve generation quality for different benchmark datasets?

**Implementation** We implement Lazy-Attention within 5K lines in Python based on PyTorch v2.7 and CUDA 12.4. Our implementation is designed to be compatible with existing Transformer architectures, allowing for easy integration into various models. We use the vLLM v0.8.5.post1 V1 (vLLM Team, 2025) framework for efficient inference and model management, using its capabilities to optimize memory usage and computational efficiency. The code is available at `https://anonymous.4open.science/r/lazy-attention-424F`.

**Models, Hardware, and Datasets** We evaluate Lazy-Attention using Tulu3-Block-FT[2] which is fine-tuned for Block-Attention (Ma et al., 2025) from Llama-3.1-Tulu-3-8B-SFT on a machine with 120GB RAM, an NVIDIA H100 96GB GPU (GH200 chipset) (NVIDIA, 2022). To demonstrate generalization, we also extend our evaluation to Llama-3.1-70B-Instruct and Qwen3-8B, and test on NVIDIA A100 and A40 GPUs (details in Appendix F). Our evaluation uses four QA benchmarks: (1) *2WikiMQA* Ho et al. (2020), which requires reading multiple paragraphs, each treated as a document in Lazy-Attention; (2) *HotpotQA* (Yang et al., 2018), a multi-hop dataset requiring reasoning across supporting documents; (3) *TriviaQA* (Joshi et al., 2017), a reading comprehension benchmark with long web-page contexts; and (4) *NarrativeQA* (Kočiský et al., 2018), where questions demand understanding long narratives such as novels and scripts.

**Baselines** We compare Lazy-Attention against the following baselines: (1) *Prompt Cache* (Gim et al., 2024), the standard RAG model using a fixed-length cached prefix; (2) *CacheBlend* (Yao et al., 2025), a masked variant of RAG that improves accuracy; (3) *Block-Attention (vLLM)* (Ma et al., 2025), a block-based mechanism for cache efficiency, re-implemented in vLLM for fairness; (4) *Prefix Caching* (Kwon et al., 2023a; vLLM Team, 2025), the standard prefix caching in vLLM.

### 4.1 HIGHER RESPONSIVENESS WITH SHORTER TTFT

We generate request streams with a controlled arrival rate (req/s) from the document and query pools, mixed four datasets. Each request references 5 documents whose per-document KV caches have been precomputed and stored in DRAM. We evaluate two regimes: *Uniform* sampling (low reuse), and *Skewed* sampling (Zipf-like with $\alpha = 2.1$, high reuse). All methods share the same batching, paging parameters, and memory budget. We report mean TTFT. We compare against all baselines and two reference bounds: *Full Reuse* (ideal lower bound) and *Full Recompute*.

---

[2]`https://huggingface.co/ldsjmdy/Tulu3-Block-FT`

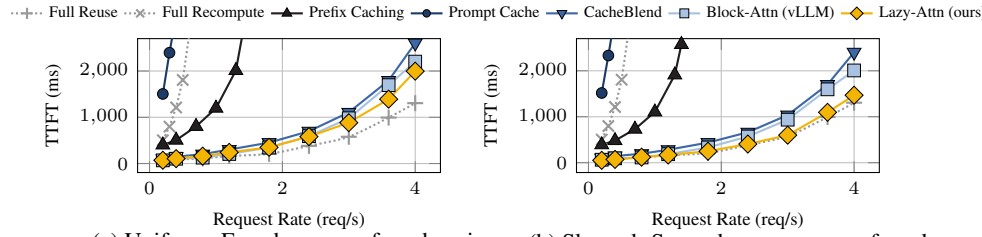

(a) Uniform: Few docs are referred again    (b) Skewed: Some docs are more referred

Figure 3: TTFT vs. request rate under different document distributions. We vary the request rate (req/s) and plot time-to-first-token (TTFT). (a) *Uniform*—few documents recur across requests (low reuse). (b) *Skewed*—a small set of documents is frequently reused (high reuse).

Table 1: VRAM cache hit ratio (%) under different KV cache budgets. A high hit ratio indicates more effective cache reuse. Our proposed approach (LazyAttention) is the most effective.

| KV Cache Mem Size ($\rightarrow$) | 1 GB | | | 5 GB | | | 10 GB | | |
|---|---|---|---|---|---|---|---|---|---|
| Document Skewness ($\rightarrow$) | Low | Mid | High | Low | Mid | High | Low | Mid | High |
| Prefix Caching | 0.7789 | 0.6214 | 0.6253 | 2.539 | 2.290 | 2.188 | 3.013 | 2.728 | 2.897 |
| RAGCache | 0.8014 | 0.6577 | 0.6449 | 2.501 | 2.267 | 2.158 | 2.910 | 2.493 | 2.562 |
| CacheBlend | 6.010 | 6.605 | 7.518 | 11.21 | 11.84 | 12.16 | 13.57 | 14.53 | 14.20 |
| Block-Attention (vLLM) | 6.037 | 6.609 | 7.200 | 11.90 | 11.50 | 12.05 | 13.66 | 13.50 | 14.38 |
| *LazyAttention (Ours)* | **7.396** | **8.891** | **9.007** | **11.52** | **13.61** | **13.70** | **18.04** | **18.56** | **20.69** |

Under *Uniform* traffic (Fig. 3a), reuse opportunities are scarce. Lazy-Attention remains competitive—closely tracking *Block-Attn* at low to moderate load and consistently outperforming *Prefix Caching*, *PromptCache*, and *CacheBlend*. This is consistent with our kernel analysis: the prefill overhead is $\frac{3}{4M}$ and the decode overhead averages $r \cdot \frac{3}{2N}$ (§3.2), which are both small when $M$ is large and the fraction $r$ of nonzero-offset tiles is low.

Under *Skewed* traffic (Fig. 3b), many requests reuse a small set of documents, often at different positions. Here Lazy-Attention achieves lower TTFT and sustains higher throughput before saturation. The gain stems from position-agnostic KV reuse: we reuse the same KV blocks across offsets without materializing copies, which avoids additional HBM footprint and paging. In contrast, methods that either recompute segments (*CacheBlend*) or expand prompts (*PromptCache*) pay extra prefill time, and materialized-position schemes (e.g., *Block-Attn*) can incur duplicate KV copies and memory pressure when offsets vary. We observe similar speedups on the larger Llama-3.1-70B model and across different hardware (Appendix F).

### 4.2 HIGHER CACHE HIT RATIOS

We vary the KV cache budget (1/5/10 GB) and document popularity skew (Low/Mid/High with $\alpha = 1.1/1.5/2.1$). The metric is VRAM cache hit ratio: the fraction of KV-block lookups served from cache without recomputation. Lazy-Attention consistently attains the highest ratios. At 1 GB, eliminating position-coupled duplicates gives clear gains (e.g., Low-skew 7.40% vs. best baseline 6.04%, +∼22%). The advantage persists at 5 GB (13.61% vs. 12.05%, +∼13%) and is most pronounced at 10 GB/High-skew (20.69% vs. 14.38%, +∼44%). Unlike Block-Attention (duplicate KV copies), CacheBlend (reconstruction overhead), and Prefix Caching (contiguous-only reuse), our single-copy design reuses documents at any offset, resulting in higher hit ratios. Therefore, Lazy-Attention has a unique advantage in RAG serving when there are hot documents and device memory, i.e., VRAM, is limited.

### 4.3 FUSED KERNEL EFFICIENCY: OUR ROTATION OVERHEAD IS NEGLIGIBLE

To isolate kernel-level costs, we construct a single long RAG request with five documents (each 4,096 tokens) and a 64-token query. The serving system uses a 2,048-token chunk budget per pass. We preload three documents' KV blocks in DRAM to emulate "hot" content and leave the remaining

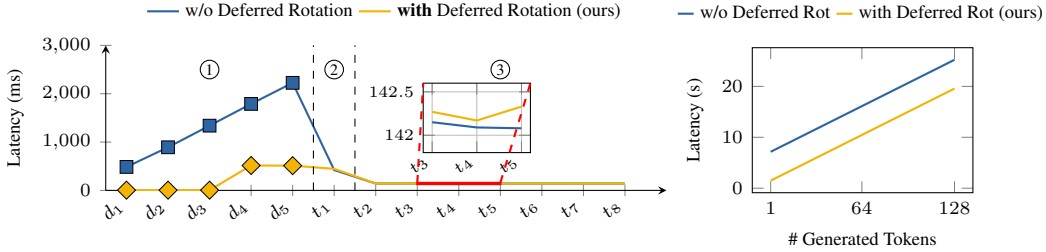

(a) Inference time over prefill (d1–d5) and decoding steps (t1–t8)  (b) Cumulative generation time

Figure 4: Overhead analysis. (a) Breakdown of overhead across phases: ① document processing, ② query pre-filling, ③ decoding. Deferred rotation adds only 0.13% overhead in decoding. (b) Accumulated latency with and without deferred rotation, where the dominant gain comes from document processing rather than token generation.

documents "cold." Common preamble is omitted. We compare two ablations: *w/o Deferred Rotation* (no position-agnostic reuse; blocks are processed in the conventional pipeline) and *with Deferred Rotation (ours)*, which reuses cached KV at arbitrary offsets and performs the necessary RoPE rotation inside the fused attention kernel. Scheduler, batching, paging, and launch parameters are identical; only the rotation path differs.

As shown in Fig. 4, (① **Document processing).** Our method turns hot documents into near-zero latency segments by reusing their KV without duplication; the baseline must recompute them, dominating total time. (② **Query prefill).** Prefill cost is similar; our kernel rotates keys once per KV tile (relative offset), adding only $\frac{3}{4M}$ fractional FLOPs per tile (Sec. 3.2). (③ **Decoding).** The two curves overlap; the zoomed inset shows our per-token overhead is only **0.13%**, consistent with the analysis $r \cdot \frac{3}{2N}$ where the fraction $r$ of nonzero-offset tiles is small, matching the analysis in Section 3.4. Fig. 4 (b) confirms that the cumulative generation gap stays flat as tokens increase—the decoding overhead does *not* accumulate, while the main gain comes from avoided document processing. We further verify that this overhead remains negligible for document lengths up to 16K and context lengths up to 128K (Appendix F).

## 4.4 GENERATION QUALITY

We evaluate Lazy-Attention following the Block-Attention setup (Ma et al., 2025), reporting exact match (EM) for QA tasks and a small human study for fluency. Unless stated, we use the same model (`Tulu3-block-ft`), prompts, and decoding parameters. We also showcase a unique application, *reordering search*, which explores diverse decoding paths. Lazy-Attention achieves EM comparable to Block-Attention, consistent with prior reports, confirming that deferred positional embedding preserves quality. Using vLLM for inference yields ∼3 lower EM than HuggingFace `transformers`, mainly due to tokenizer/version mismatches affecting tokenization and precision. We also evaluate a long-form literature review task, showing Lazy-Attention accelerates generative tasks beyond simple QA (Appendix F).

Table 2: Question-answer accuracy for various benchmark datasets. *Exact match* scores are reported. Our Lazy-Attention performs the identical computations as Block-Attention, achieving nearly identical scores, where slight differences are due to tokenization and limited precision in floating-point operations. That is, Lazy-Attention significantly reduces TTFT and increases the reuse opportunities (as reported before) with negligible accuracy loss.

| Dataset (↓) | Full-Attn | CacheBlend | Block-Attn | *Block-Attn (vLLM)* | *Lazy-Attn (ours)* |
|---|---|---|---|---|---|
| 2WikiMQA | 73.6 | 71.1 | 72.2 | 71.4 | 70.7 |
| TriviaQA | 75.2 | 69.2 | 72.3 | 72.1 | 73.0 |
| NarrativeQA | 62.2 | 60.1 | 60.4 | 61.0 | 59.7 |
| HotpotQA | 76.2 | 69.7 | 75.1 | 72.5 | 73.3 |
| Average | 71.8 | 67.5 | 71.2 | 69.3 | 69.2 |

## 5 CONCLUSION

In this paper, we proposed Lazy-Attention, a novel deferred positional encoding mechanism for Transformer models. Our approach decouples positional encoding from the KV cache, allowing for more efficient caching and improved performance in retrieval-augmented generation tasks. We demonstrated the effectiveness of our method through extensive experiments on various datasets, showing significant improvements in TTFT, cache hit ratio, and overall model performance. Our findings suggest that Lazy-Attention is a promising solution for enhancing the efficiency and accuracy of Transformer models in long-context tasks.

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

## A    RELATED WORK

We contextualize Lazy-Attention within the landscape of KV reuse and efficient attention, distinguishing it from prior works based on reuse flexibility, memory overhead, and architectural requirements.

**Distinction from RAG-oriented Caching (RAGCache, Mooncake)** Systems like RAG-Cache (Jin et al., 2024) and Mooncake (Qin et al., 2024) rely on Prefix Caching. They optimize cache management (e.g., eviction policies like PGDSF) but are limited by the underlying prefix-match requirement. If a document appears in a different position (e.g., shifted by a system prompt), these systems cannot reuse the cached KV and must re-generate the states from scratch and store a duplicate copy in HBM. Lazy-Attention is complementary to these systems: while they focus on *what* to keep, we optimize *how* it is used, decoupling position from storage to allow a single physical copy to serve logically distinct requests.

**Distinction from Re-encoding Methods (Block-Attn, TurboRAG)** This is a critical distinction. While TurboRAG (Lu et al., 2025) and Block-Attention (Ma et al., 2025) achieve position flexibility, they are fundamentally limited by memory materialization. They must write the re-encoded KV states into a new memory buffer to perform attention. This "Read-Modify-Write" pattern consumes HBM capacity and memory bandwidth. In contrast, Lazy-Attention fuses the re-encoding into the attention kernel on-the-fly. This eliminates the need for duplicate memory copies and avoids the bandwidth overhead of writing back to HBM, enabling Zero-Copy reuse.

**Distinction from MLA / TransMLA** MLA (DeepSeek-AI et al., 2024; DeepSeek-AI et al., 2025) and TransMLA (Meng et al., 2025) utilize a "Decoupled RoPE" strategy, but their primary objective is KV compression (via low-rank projection), not position-agnostic reuse. In MLA, RoPE is applied to a subset of dimensions to preserve the low-rank structure, but the attention computation still relies on absolute positional alignment. Consequently, MLA-based systems remain restricted to Prefix Caching paradigms and cannot reuse KV blocks when their positions change within the prompt. In contrast, Lazy-Attention enables zero-copy sharing of KV blocks at arbitrary positions on standard architectures.

**Efficient attention and kernel fusion** FlashAttention (Dao et al., 2022; Dao, 2024; Shah et al., 2024), FlexAttention (Dong et al., 2024), and memory/paging systems such as FlexGen (Sheng et al., 2023) reduce traffic via IO-aware tiling and fusion. Our kernels follow the same IO-aware principles: in the prefilling phase, we add only $1/(2N)$ extra reads per KV tile; in the decoding phase we add essentially *no* extra inner-loop loads (packed metadata; in-register rotation), making our method complementary to these optimizations.

# B  EXPERIMENTS DETAILS

## B.1  DETAILS OF COMPARED METHODS

- **Full Recomputation**: vLLM with `enable_prefix_caching=False`.
- **Prefix Caching**: vLLM with `enable_prefix_caching=True`.
- **Full Reuse**: The KV caches from individual documents are concatenated and used directly as the KV cache for the concatenated documents.
- **Block-Attention (Official)** (Ma et al., 2025): Evaluated using the official implementation[3].
- **Block-Attention (vLLM)**: We reimplemented Block-Attention (Ma et al., 2025) within the vLLM v1 engine. If a document is found in the cache but its position does not align and its reference count is zero (i.e., no other request is using it), rotation can be applied directly to adjust the positional encoding. Otherwise, new memory is allocated, the blocks are copied, and then rotated.
- **CacheBlend** (Yao et al., 2025): Evaluated using the official LMCache examples[4]. Multiple documents are concatenated with `blend_special_str` to form the prompt, which is then processed by a vLLM instance integrated with LMCache.
- **Prompt Cache** (Gim et al., 2024): Evaluated using the official implementation[5].

# C  REPRODUCIBILITY STATEMENT

We release anonymized source code and experiment scripts at `https://anonymous.4open.science/r/lazy-attention-424F`. Our repository contains instructions for setting up the environment, launching experiments, and reproducing all figures and tables in the paper. The detailed evaluation setup is described in Section 4 and our codebase `https://anonymous.4open.science/r/lazy-attention-424F`. Together, these resources ensure that our results can be replicated and extended by the research community.

# D  ETHICS STATEMENT

This work does not involve human subjects, proprietary datasets, or personally identifiable information. Our contributions focus on system design for special attention computation, which we call Lazy-Attention. Potential risks are those inherent to large language models, such as misuse for generating harmful content, which are unchanged by our methods. Finally, we used an LLM only for grammar checking in writing this paper.

# E  THE USE OF LARGE LANGUAGE MODELS

We used large language models solely to polish wording and grammar (clarity, tone). No technical content (algorithms, experiments, figures, analyses) was generated by LLMs. All suggested edits were manually reviewed and approved by the authors.

---

[3]`https://github.com/TemporaryLoRA/Block-Attention`

[4]`https://github.com/LMCache/LMCache`

[5]`https://github.com/yale-sys/prompt-cache`

# F    ADDITIONAL EXPERIMENTAL RESULTS

## F.1    GENERALIZATION TO LARGER MODELS (70B)

We evaluate Lazy-Attention on Llama-3.1-70B-Instruct deployed on a node with 4×H100 GPUs (Tensor Parallelism = 4), under the same RAG QA workload as Section 4.1 (5 retrieved documents, 1 request/s). Table 3 shows that Lazy-Attention achieves a 5.2× TTFT speedup over standard RAG serving. Moreover, the gap between Lazy-Attention and CacheBlend widens from 1.43× on 8B to 1.53× on 70B, because larger models have substantially larger KV states (more layers and wider dimensions) and are therefore more memory-bandwidth bound. By avoiding materialization of position-shifted KV blocks, our zero-copy design saves HBM bandwidth, and this advantage amplifies as model size increases.

Table 3: TTFT on RAG QA for 8B vs 70B models.

| Model | Method | TTFT (ms) | Speedup |
|---|---|---|---|
| Tulu3-Block-FT 8B | Standard RAG | 1196.8 | 1.0× |
| | CacheBlend | 274.8 | 4.4× |
| | **Lazy-Attention** | **191.7** | **6.2×** |
| Llama-3.1-70B (TP=4) | Standard RAG | 1253.0 | 1.0× |
| | CacheBlend | 365.5 | 3.4× |
| | **Lazy-Attention** | **238.4** | **5.2×** |

## F.2    PERFORMANCE ON DIFFERENT HARDWARE (A100, A40)

To test robustness across hardware generations, we further evaluate Lazy-Attention on NVIDIA A100 (40GB) and A40 (48GB) GPUs, which provide substantially lower memory bandwidth than H100. As summarized in Table 4, the relative speedup of Lazy-Attention over CacheBlend grows as memory bandwidth decreases (from 1.43× on H100 to 1.70× on A40). This trend follows from the fact that Block-Attention and CacheBlend use a "Read-Modify-Write" pattern to materialize position-adjusted KV blocks, effectively doubling HBM traffic. On bandwidth-constrained GPUs like A40, this extra traffic becomes the bottleneck. In contrast, Lazy-Attention keeps KV accesses strictly read-only and avoids this bandwidth penalty.

Table 4: TTFT Comparison under Uniform Sampled Documents across different GPUs.

| Hardware | Method | TTFT (ms) | Speedup (vs Prefix) |
|---|---|---|---|
| H100 | Prefix Caching | 1196.8 | 1.0× |
| (High BW) | CacheBlend | 274.8 | 4.3× |
| | **Lazy-Attention** | **191.7** | **6.2×** |
| A100 | Prefix Caching | 2580.4 | 1.0× |
| (Mid BW) | CacheBlend | 565.3 | 4.5× |
| | **Lazy-Attention** | **372.5** | **6.9×** |
| A40 | Prefix Caching | 4150.5 | 1.0× |
| (Low BW) | CacheBlend | 1150.6 | 3.6× |
| | **Lazy-Attention** | **675.2** | **6.1×** |

## F.3    VERSATILITY IN TASKS: LONG-FORM LITERATURE REVIEW

To assess versatility beyond factoid-style QA, we construct a Long-form Literature Review task: the model receives 5 ArXiv papers (converted to text, ∼8K tokens each) and is asked to produce a 1024-token literature review. This setting stresses the system on both long-context prefilling and subsequent decoding. As shown in Table 5, Lazy-Attention improves end-to-end latency by 1.7×, indicating that the benefits of our design extend from TTFT to overall response time in more realistic generative workloads.

Table 5: Long-form literature review latency (Tulu3-Block-FT 8B, H100).

| Method | End-to-end Latency (s) | Speedup |
|---|---|---|
| Standard RAG | 38.0 | 1.0× |
| **Lazy-Attention** | **22.4** | **1.7×** |

## F.4 LONG-CONTEXT SCALABILITY AND NUMERICAL STABILITY

We next study scalability to longer documents (up to 16K tokens per document) and numerical stability at very long sequence lengths (up to 128K tokens). Table 6 shows that the TTFT speedup of Lazy-Attention remains roughly constant as document length increases, indicating that our fixed tiling strategy stays efficient even for long contexts. For stability, Lazy-Attention applies stateless on-the-fly rotations using absolute token indices ($m\theta_i$), instead of iteratively updating the rotation state. This design prevents error accumulation across layers or timesteps. Table 7 shows that the maximum difference in attention logits versus standard attention remains below $10^{-5}$ even at 128K tokens, confirming that our kernel is numerically stable.

Table 6: Performance scaling on varying document lengths (5 docs per request).

| Doc length | Standard TTFT | Lazy-Attn TTFT | Speedup |
|---|---|---|---|
| 4K | 7.152 s | 1.487 s | 4.81× |
| 8K | 14.133 s | 2.919 s | 4.84× |
| 16K | 28.446 s | 5.721 s | 4.97× |

Table 7: Consistency with standard attention (H100, 128K tokens).

| Seq Length | Max abs diff (logits) | Max relative diff (logits) |
|---|---|---|
| 128K | 3.7457e-5 | 5.4829e-7 |

## F.5 GENERALIZATION TO OTHER ARCHITECTURES AND METHODS

We also apply Lazy-Attention to Qwen3-8B (Yang et al., 2025), which uses a different RoPE implementation, and combine it with Lego-Link0 (Hu et al., 2025), a training-free cache reuse strategy. Table 8 reports consistent speedups of about 6.3× across these settings, indicating that our kernel is robust to architectural changes and is complementary to both fine-tuned and training-free reuse methods.

## F.6 SENSITIVITY ANALYSIS OF TILING PARAMETERS

We perform a sensitivity study over different prefill tile sizes $M$ while fixing the decode tile size at $N = 64$. As summarized in Table 9, normalized throughput varies by at most 3% across the tested values of $M$. This weak dependence on $M$ suggests that our default configuration ($M = 128$) is near-optimal and robust across a wide range of document lengths.

## F.7 EXTENDED CACHE HIT RATIO ANALYSIS

Finally, we extend the cache hit ratio analysis to larger VRAM budgets (50GB and effectively unlimited). Table 10 shows that Lazy-Attention consistently attains the highest hit ratios across all budgets and skewness levels, even when memory is plentiful. This improvement arises because our zero-copy kernel allows each cached document to cover more effective positions, thereby increasing the coverage of hot documents without increasing memory footprint.

Table 8: Generalization on Qwen3-8B and Lego-Link0.

| Model / Method | Standard TTFT | Lazy-Attn TTFT | Speedup |
|---|---|---|---|
| Tulu3-Block-FT 8B | 1196.8 ms | 191.7 ms | 6.2× |
| Qwen3-8B | 1277.4 ms | 201.2 ms | 6.35× |
| Lego-Link0 + Llama-3.1-8B | 1195.6 ms | 191.3 ms | 6.2× |

Table 9: Normalized throughput vs. prefill tile size $M$ (fixed $N = 64$).

| Doc Length (tokens) | M = 64 | M = 128 (Ours) | M = 256 |
|---|---|---|---|
| Short (256) | 1.00× | 0.99× | 0.97× |
| Medium (4K) | 0.99× | 1.00× | 0.99× |
| Long (16K) | 0.98× | 1.00× | 1.00× |

Table 10: VRAM cache hit ratio (%) with large memory budgets.

| Mem Size | Skewness | Prefix Caching | RAGCache | CacheBlend | Block-Attn | **Lazy-Attn** |
|---|---|---|---|---|---|---|
| 50GB | Low | 4.70 | 4.52 | 18.80 | 19.32 | **23.43** |
| | High | 4.47 | 4.01 | 18.85 | 19.29 | **26.71** |
| No Limit | Low | 4.94 | 4.76 | 19.61 | 20.15 | **24.48** |
| | High | 4.71 | 4.22 | 19.57 | 20.06 | **27.87** |

