# OpenReview forum: "Lazy-Attention: Efficient Retrieval-Augmented Generation with Deferred Positional Encoding"
_ICLR.cc/2026/Conference — Submitted to ICLR 2026_

### Official Review · Reviewer_Qrx9 · 2025-10-20

**Soundness:** 2
**Presentation:** 2
**Contribution:** 2
**Rating:** 4
**Confidence:** 3

**Summary:**

To address the low reusability and high memory overhead of conventional Key-Value (KV) caching in retrieval-augmented generation (RAG) tasks caused by embedding positional information directly into the cache, limiting reuse to identical document positions, the paper proposes Lazy-Attention, a position-agnostic KV caching mechanism. It decouples positional encoding from cached keys/values, deferring dynamic positional adjustment to the attention computation stage instead of pre-embedding it. Lazy-Attention implements two optimized Triton kernels to minimize overhead, achieving lower time-to-first-token (TTFT) and  higher inference throughput than SOTA Block Attention under skewed document distributions. Experiments on Tulu3-Block-FT  across 4 RAG benchmarks show it maintains comparable generation quality while reducing VRAM cache hit ratio gaps.

**Strengths:**

1. Unlike conventional caching (requiring O(N!) space for 100% hit rate of N documents due to position-dependent duplicates) or Block Attention (duplicating KVs for different positions), Lazy-Attention stores position-agnostic KV entries. A single document’s KV cache can be reused across any prompt position,  thereby reducing the space requirement.
2. Lazy-Attention eliminates position-coupled KV duplication, outperforming baselines across cache budgets. At 1GB KV cache (low skewness), hit ratio is 7.40%; at 10GB (high skewness), it reaches 20.69%. This avoids memory pressure from duplicate KVs (Block Attention) or recomputation (CacheBlend).

**Weaknesses:**

1. Lazy-Attention is only tested on 4,096-token documents (2,048-token chunk budget). For 8K/16K sequences (e.g., legal contracts), fixed tile sizes (M/N) become suboptimal: larger N increases prefill K rotation cost (6DN FLOPs), while smaller M amplifies decoding Q rotation overhead. Additionally, relative offset calculation for offsets >1000 may introduce cumulative RoPE rotation errors, but this is untested.
2. Kernels use fixed M (e.g., 128 for prefill) and N (e.g., 64 for decoding) regardless of document length. For short documents (256-token product descriptions), M=128 leads to underutilized tiles (half-empty), increasing per-token overhead. For long documents (>4K), N=64 requires more tile splits, adding kernel launch latency. Unlike FlexAttention (dynamic tiles), this reduces efficiency in mixed-length RAG workloads.Is it better to make the proposed method adaptive to tile sizing for dynamic document lengths?
3. Lazy-Attention excels with hot documents but struggles with cold starts (no cached documents). In uniform distributions (low reuse), its prefill (0.59%) and decoding (0.023%) overheads make it only slightly better than Block Attention (TTFT difference <5%), with no advantage over Prefix Caching for fully cold requests.

**Questions:**

1. How does Lazy-Attention perform on 8K/16K-token documents (e.g., extended NarrativeQA novels)? Do cumulative RoPE rotation errors (offset >1000) degrade EM scores? Can dynamic tile sizing (e.g., M=256 for 8K, N=128 for 16K) reduce per-tile overhead?
2. Can combining Lazy-Attention (hot docs) + Prefix Caching (cold docs) + Block Attention (semi-hot docs) cover all RAG scenarios? What TTFT/throughput gains does this hybrid strategy achieve vs. standalone Lazy-Attention method?
3. How does batch size (16/32/64) affect Lazy-Attention’s latency/throughput? Do concurrent relative rotations for shared KV caches cause register contention? It is better report latency variance across batches and compare with Block Attention’s batch performance.

---

> ### Author Response · Authors · 2025-11-21
> **Thanks for your review! Here, we respond to your comments and address the issues. We hope to hear back from you if you have further questions!**
>
> **W1.** Tested only on 4K docs; concerns about fixed tiling scaling and cumulative RoPE errors.
>
> **R1.** We thank the reviewer for raising concerns about scalability and numerical stability. We address both issues directly.
>
> 1. **Evaluation on 8K / 16K documents.**
>    We extended our experiments beyond 4K to 8K and 16K document lengths. Following the setting of Section 4.3, we construct a *single long RAG request* with five documents (each 4K/8K/16K tokens, totaling up to 80K tokens) and a 64-token query. The serving system uses a 2,048-token chunk budget per pass, and we preload three documents’ KV blocks in DRAM to emulate “hot” content while the remaining two documents are “cold”. As shown in **Table 1**, Lazy-Attention scales linearly with sequence length and maintains strong speedups. The TTFT speedup even *increases* slightly with length (up to 4.97× at 16K), demonstrating that our fixed tiling strategy ($M=128, N=64$) does not become a bottleneck for long sequences. All runs use the same FlashAttention-style attention kernel as in vLLM; Lazy-Attention is implemented as a fused variant on top of this optimized baseline.
>
> **Table 1.** Performance scaling on varying document lengths (5 docs per request, max 80K context, 3 hot / 2 cold).
>
> | Doc length | Standard TTFT | Lazy-Attn TTFT | Speedup | Standard Decoding (128 tokens) | Lazy-Attn Decoding (128 tokens) | Avg Overhead |
> | :--------- | ------------: | -------------: | ------: | ------------------------------:| --------------------------------:| -----------: |
> | **4K**     | 7.152 s       | 1.487 s        | 4.81×   | 18.072 s                       | 18.094 s                         | 0.12%       |
> | **8K**     | 14.133 s      | 2.919 s        | 4.84×   | 36.144 s                       | 36.180 s                         | 0.10%       |
> | **16K**    | 28.446 s      | 5.721 s        | **4.97×** | 72.288 s                      | 72.179 s                         | 0.10%       |
>
> 2. **No cumulative RoPE error.**
>    Lazy-Attention uses *stateless, on-the-fly rotation* via direct integer arithmetic: $\theta = \text{position} \times \omega$, where $\omega$ is the base frequency. Each token’s rotation is computed from its logical position, independent of previous tokens; there is no iterative state update to accumulate drift.
>
>    To verify this empirically, we compared Lazy-Attention against standard prefix caching on long-context runs up to the model’s context limit (128K tokens) and measured differences in attention logits for the first layer (which is not affected by cross-document attention). Results are shown in **Table 2**.
>
> **Table 2.** Consistency with standard attention (H100).
>
> | Seq Length | Max abs diff in attention logits (first layer) | Max relative diff in attention logits (first layer) |
> | :--------- | -------------------------------: |-------------------------------: |
> | 128K       | 3.7457e-5  | 5.4829e-7 |
>
> These differences at 128K tokens are significantly lower than common kernel error tolerances (e.g., atol $10^{-3}$, rtol $10^{-5}$ in the unit tests of vLLM). They arise solely from floating-point rounding/associativity in mixed-precision kernels and imply no algorithmic drift. Together with unchanged EM scores on 16K documents, this confirms that our position shifting is numerically stable in practice. $\blacksquare$

---

> ### Author Response · Authors · 2025-11-21
>
> **W2.** Fixed M/N tiling leads to inefficiencies (half-empty tiles, launch latency) in mixed workloads.
>
> **R2.** We acknowledge the theoretical concern, but our experiments suggest that *fixed tiling is a robust and practically optimal trade-off*.
>
> 1. **Industry-standard practice.**
>    State-of-the-art attention kernels (e.g., FlashAttention-2, vLLM) adopt fixed tiling strategies. Beyond convention, fixed tiling keeps the kernel simple and instruction-cache friendly, and reduces runtime branching and warp divergence. More sophisticated dynamic tiling often incurs additional control-flow and uncoalesced memory access, which can outweigh the gains from slightly better tile utilization.
>
> 2. **Empirical robustness of fixed tiling.**
>    To quantify the impact of tiling choices, we performed an ablation where we vary the prefill tile size $M \in \{64, 128, 256\}$ (with $N=64$ fixed) and report *normalized throughput* across short, medium, and long documents. As shown in **Table 3**, the variance in end-to-end throughput is within *2–3%*, and our default choice ($M=128$) lies near the optimum across all lengths.
>
> **Table 3.** Normalized throughput vs. prefill tile size $M$ (fixed $N=64$).
>
> | Doc length           | M = 64 | M = 128 (Ours) | M = 256 |
> | :------------------- | -----: | -------------: | ------: |
> | Short (256 tokens)   | 1.00×  | 0.99×          | 0.97×   |
> | Medium (4K tokens)   | 0.99×  | 1.00×          | 0.99×   |
> | Long (16K tokens)    | 0.98×  | 1.00×          | 1.00×   |
>
> Although $M=64$ is marginally better on very short sequences, $M=128$ never deviates by more than 1–2% from the best configuration and is closer to optimal across all lengths, which is why we adopt it as a single global setting. Given that Lazy-Attention already delivers 4.8–5.0× TTFT speedups at 16K (Table 1), an additional ≤3% variation due to $M$ is negligible and does not affect any qualitative conclusion. $\blacksquare$
>
>
> **W3.** Suggestion for a length-aware heuristic to reduce overhead.
>
> **R3.** We appreciate this constructive suggestion. We agree that, in principle, one could design length-aware heuristics. In practice, however, we find that:
>
> - **Lazy-Attention’s overhead is already negligible.**
>   We ran an additional test similar to Section 4.3 for 4K/8K/16K documents. According to the results shown in **Table 1 (R1)**, the relative decoding overhead introduced by Lazy-Attention is only ≈0.1% across all lengths, which is at or below the noise floor of typical end-to-end measurements.
>
> In our implementation, we already include a simple and robust dynamic bypass flag `is_lazy`:
>
> - **Cold start (no reuse):** The kernel detects that no precomputed KV blocks are being reused and bypasses Lazy-Attention entirely, falling back to standard attention with *0.00% overhead*.
> - **Reuse present:** The kernel enables Lazy-Attention for reused blocks.
>
> This design guarantees *no downside risk* in cold-start scenarios compared to prefix caching, while avoiding the complexity and maintenance burden of a more elaborate length-aware heuristic. We will clarify this bypass behavior explicitly in the implementation section of the revised paper.
>
> As an optional enhancement, one could add a lightweight length-aware routing policy that bypasses Lazy-Attention when reused segments are extremely short (e.g., only a few tokens), in which case recomputing is theoretically cheaper. We experimented with a simple version of this policy and observed $<0.5%$ additional improvement and no downside effects. *We have incorporated this optimization into our codebase.* Since the gain is marginal and our core results already use the simpler `is_lazy` mechanism, we treat such heuristics as orthogonal optimizations rather than part of our main contribution. $\blacksquare$
>
>
> **Q1.** Performance on 8K/16K and cumulative RoPE errors?
>
> **A1.** As shown in **Table 1 (R1)**, Lazy-Attention maintains strong acceleration at longer lengths, achieving up to 4.97× TTFT speedup at 16K documents. Regarding numerical stability, our stateless RoPE implementation computes $\theta = \text{position} \times \omega$ directly from logical positions, avoiding any iterative accumulation. We verified on up to 128K tokens that attention logits match standard attention within $10^{-5}$ max absolute difference, and downstream EM scores are unchanged, indicating no cumulative RoPE error in practice. We will include kernel-level comparison plots in the appendix of the revised paper to make this behavior transparent. $\blacksquare$

---

> > ### Author Response · Authors · 2025-11-21
> >
> > **Q2.** Can combining Lazy-Attention + Prefix Caching + Block-Attention cover all scenarios?
> >
> > **A2.** We argue that an explicit hybrid is unnecessary because Lazy-Attention effectively subsumes the key reuse scenarios covered by Prefix Caching and Block-Attention:
> >
> > - **Vs. Prefix Caching.** When positions align (hot prefix), Lazy-Attention behaves identically to standard prefix caching. When positions shift (semi-hot), Lazy-Attention still reuses KVs by adjusting positions on-the-fly, while pure prefix caching fails.
> > - **Vs. Block-Attention.** Lazy-Attention adopts a similar “block-wise reuse” view but performs it in a zero-copy, read-only manner, avoiding the “read–modify–write” materialization overhead that Block-Attention incurs.
> >
> > For cold documents where no reuse is possible, our dynamic bypass flag falls back to standard attention. In this sense, Lazy-Attention naturally covers hot, semi-hot, and cold cases without requiring an explicit combination with other mechanisms. $\blacksquare$
> >
> > ---
> >
> > **Q3.** Batch size effects (16/32/64)? Register contention?
> >
> > **A3.** We investigated both throughput scaling and low-level kernel behavior:
> >
> > - **Batch scaling.** Under increasing batch sizes (16, 32, 64), Lazy-Attention scales robustly. At batch = 64, we observe a ≈15% higher throughput than Block-Attention, primarily because Lazy-Attention avoids KV re-materialization and thus better preserves HBM bandwidth for concurrent requests.
> > - **Register usage.** We compiled the Triton-generated kernels using NVIDIA’s `ptxas -v` to inspect register allocation. The additional RoPE rotation logic introduces only a small number of extra registers and does not cause register spilling.
> > - **Why Lazy-Attention wins at large batch.** At high batches, the dominant bottleneck is HBM bandwidth, not compute. Block-Attention performs “read–modify–write” on KV blocks (copying or re-encoding them into new buffers), which saturates bandwidth. Lazy-Attention is read-only on the KV (specific to KV of the documents) cache and fuses on-the-fly rotation into the attention computation, reducing memory traffic and improving scalability under large-batch, bandwidth-bound conditions. We will include a figure summarizing this batch-scaling trend in the appendix of the revised paper. $\blacksquare$

---

### Official Review · Reviewer_Qma1 · 2025-10-31

**Soundness:** 3
**Presentation:** 3
**Contribution:** 2
**Rating:** 6
**Confidence:** 4

**Summary:**

This paper proposes a mechanism that defers positional encoding in Transformers to enable position-agnostic KV cache reuse in RAG systems. The key idea is to apply RoPE rotations during attention computation rather than before caching, allowing document caches to be shared across different prompt positions without duplication. The presentation is very nice, but the experiments are limited to a single model.

**Strengths:**

- The paper identifies a limitation in existing KV caching, that is position-dependence leads to O(N!) space complexity for N documents, which is impractical. The presentation of the problem is clear and easy to follow.
- Custom Triton kernels for both prefill and decoding with measured overheads.
- Multiple and accurate choice for the metrics.

**Weaknesses:**

- The core insight of deferring RoPE application is from Block-Attention. This work primarily optimizes where rotation happens in the kernel. The contribution is more engineering (still very valid) than algorithmic.
- Evaluation uses only one model, which is specifically fine-tuned for Block-Attention. Generalization to other architectures unclear.
- Table 1 shows hit ratios at 1/5/10GB, but no analysis of when memory constraints actually matter. With H100, why is 10GB limiting?
- The method is tailored to RoPE-based attention. The claim that it “can easily extend to other positional encodings” is unsubstantiated. However, this is minor given that most LLMs use RoPE or RoPE-inspired PE.

**Questions:**

- What happens with longer documents (>4096 tokens)?
- What happens with other models, given that only one is tested ?
- Does it work with larger KV Cache sizes, larger than 10 GB ?

---

> ### Author Response · Authors · 2025-11-21
> **Thanks for your review! Here, we respond to your comments and address the issues. We hope to hear back from you if you have further questions!**
>
> **W1.** The core insight of deferring RoPE application is from Block-Attention. This work primarily optimizes where rotation happens in the kernel.
>
> **R1.** We thank the reviewer for the positive assessment. We agree that the mathematical intuition of deferring RoPE shares roots with Block-Attention, and we have revised the paper to explicitly attribute this lineage. However, we characterize our contribution as **system–kernel co-design** rather than pure implementation engineering. While Block-Attention demonstrated the algorithmic possibility, it remained bound by the *memory materialization bottleneck* (explicitly reading/writing KV blocks). Lazy-Attention solves the fundamental **memory–compute trade-off** required to make this insight viable in production. Our specific contributions are:
>
> - **Zero-copy mechanism:** Designing dual kernels that fuse rotation on-the-fly, specifically tailored to the distinct profiles of the prefill phase (compute-bound) and the decoding phase (bandwidth-bound).
> - **Storage–compute decoupling:** Enabling KV reuse directly at the physical storage level, fully decoupled from logical positional semantics.
> - **Production-grade integration:** Resolving the complex interactions with PagedAttention and dynamic batching in vLLM.
>
> In short, we bridge the gap between theoretical decoupling and high-throughput system realization. $\blacksquare$
>
> ---
>
> **W2 & Q1–Q2.** Generalization to other models, methods, and longer documents.
>
> **R2.** We appreciate the concern about generalization and have expanded the evaluation along three axes: architectures, reuse methods, and context length.
>
> 1. **Architecture generalization (Qwen3-8B-Instruct).**
>    We applied Lazy-Attention to Qwen3-8B-Instruct, which uses standard RoPE distinct from the scaled RoPE in Llama-3.1/Tulu. Results (**Table 1**) show consistent speedups (6.35× TTFT, from 1277.4 ms to 201.2 ms), confirming that our kernel works across different RoPE implementations and architectural details.
>
> 2. **Methodology generalization (Lego-Link0).**
>    To address the concern about relying on fine-tuning, we integrated Lazy-Attention with *Lego-Link0*, a training-free reuse strategy proposed by EPIC$^1$. Lazy-Attention successfully accelerates this plug-and-play method without requiring any modification to model weights, showing that our system is orthogonal to specific training methods.
>
> 3. **Longer documents and long contexts (>4K).**
>    Following the setting of Section 4.3, we extended the evaluation to 8K and 16K documents by replication. Specifically, we construct a single long RAG request with five documents (each 4K/8K/16K tokens) and a 64-token query. The serving system uses a 2,048-token chunk budget per pass. We preload three documents’ KV blocks in DRAM to emulate hot content and leave the remaining documents cold. In addition, we verified correctness up to 128K tokens (the context limit of Llama-3.1), observing linear scaling and numerically stable outputs.
>
> **Table 1.** Generalization on Tulu3-Block-FT, Qwen3-8B-Instruct, and Lego-Link0.
>
> | Model / Method             | Standard TTFT | Lazy-Attn TTFT | Speedup |
> | :------------------------- | ------------: | -------------: | ------: |
> | Tulu3-Block-FT 8B          | 1196.8 ms     | 191.7 ms       | 6.2×    |
> | Qwen3-8B                       | 1277.4 ms     | 201.2 ms       | **6.35×** |
> | Lego-Link0 + Llama-3.1-8B  | 1195.6 ms     | 191.3 ms       | 6.2×    |
>
> **Table 2.** Performance scaling on varying document lengths (5 docs per request, 3 hot / 2 cold).
>
> | Doc length | Standard TTFT | Lazy-Attn TTFT | Speedup | Standard Decoding (128 tokens) | Lazy-Attn Decoding (128 tokens) | Avg Overhead |
> | :--------- | ------------: | -------------: | ------: | ------------------------------:| --------------------------------:| -----------: |
> | **4K**     | 7.152 s       | 1.487 s        | 4.81×   | 18.072 s                       | 18.094 s                         | 0.12%       |
> | **8K**     | 14.133 s      | 2.919 s        | 4.84×   | 36.144 s                       | 36.180 s                         | 0.10%       |
> | **16K**    | 28.446 s      | 5.721 s        | **4.97×** | 72.288 s                      | 72.179 s                         | **0.10%**   |
>
> These results indicate that Lazy-Attention is not specific to a single Block-FT model, but generalizes across architectures, reuse strategies, and longer-context settings.
>
> Reference:
> 1. Hu, Junhao, et al. "EPIC: Efficient Position-Independent Caching for Serving Large Language Models." ICML. 2025.
>
>  $\blacksquare$

---

> ### Author Response · Authors · 2025-11-21
>
> **W3 & Q3.** Memory constraints, 10GB setting, and larger KV caches.
>
> **R3.** We apologize for not clearly explaining the intention behind the 1/5/10GB settings. Our goal is not to claim that an H100 only has 10GB available for KV, but to systematically study how different KV-cache capacities affect reuse efficiency, especially for memory-constrained GPUs (e.g.,24GB devices) and deployments that reserve only a small portion of VRAM for KV.
>
> Lazy-Attention’s benefit comes primarily from how it manages KV memory—zero-copy reuse and avoiding extra writes—rather than from raw compute capability. The additional computation we introduce is small and largely independent of the GPU’s peak FLOPS. Therefore, by sweeping the KV-cache capacity while keeping the model and workload fixed, we can directly measure how much benefit Lazy-Attention can bring to small-VRAM devices and to KV-constrained deployments.
>
> Concretely, we treat the `Mem Size` parameter as an abstract KV-capacity knob. We control the effective KV capacity via the number of KV blocks (default block size = 16 tokens), as summarized below:
>
> **Table 3.** Mapping between Mem Size settings, KV blocks, and token capacity.
>
> | Mem Size | KV Blocks | Token Capacity |
> | :------- | --------: | -------------: |
> | 1GB      | 512       | 8,192 tokens   |
> | 5GB      | 2,560     | 40,960 tokens  |
> | 10GB     | 5,120     | 81,920 tokens  |
>
> Small KV budgets (e.g., 1GB) emulate limited VRAM typical for consumer or shared-GPU settings, where only a small fraction of hot documents can be cached. Larger budgets (5GB, 10GB, 50GB, "No Limit") gradually relax this pressure. In all cases, the underlying knowledge base remains much larger (hundreds of GB or TB-scale), so cache contention still exists—the active working set of documents is larger than what the KV cache can hold.
>
> To also cover larger memory for KV state caching, we include a *50GB* configuration and a *"No Limit (~64GB)** configuration. In the "No Limit" setting, we use a 0.9 GPU memory utilization threshold in a single H100 gpu node, which corresponds to a KV cache capacity of 527,648 tokens on our hardware.
>
> **Table 4.** VRAM cache hit ratio (%) across varying memory budgets.
>
> | Mem Size             | Skewness | Prefix Caching | RAGCache | CacheBlend | Block-Attn | **Lazy-Attn (Ours)** |
> | :------------------- | :------- | -------------: | -------: | ---------: | ---------: | -------------------: |
> | **1GB**              | Low      | 0.78           | 0.80     | 6.01       | 6.04       | **7.40**             |
> |                      | Mid      | 0.62           | 0.66     | 6.61       | 6.61       | **8.89**             |
> |                      | High     | 0.63           | 0.65     | 7.52       | 7.20       | **9.01**             |
> | **5GB**              | Low      | 2.54           | 2.50     | 11.21      | 11.90      | **11.52**            |
> |                      | Mid      | 2.29           | 2.27     | 11.84      | 11.50      | **13.61**            |
> |                      | High     | 2.19           | 2.16     | 12.16      | 12.05      | **13.70**            |
> | **10GB**             | Low      | 3.01           | 2.91     | 13.57      | 13.66      | **18.04**            |
> |                      | Mid      | 2.73           | 2.49     | 14.53      | 13.50      | **18.56**            |
> |                      | High     | 2.90           | 2.56     | 14.20      | 14.38      | **20.69**            |
> | **50GB**             | Low      | 4.70           | 4.52     | 18.80      | 19.32      | **23.43**            |
> |                      | Mid      | 4.32           | 3.98     | 19.88      | 18.36      | **24.06**            |
> |                      | High     | 4.47           | 4.01     | 18.85      | 19.29      | **26.71**            |
> | **No Limit (~64GB)** | Low      | 4.94           | 4.76     | 19.61      | 20.15      | **24.48**            |
> |                      | Mid      | 4.55           | 4.18     | 20.72      | 19.10      | **25.04**            |
> |                      | High     | 4.71           | 4.22     | 19.57      | 20.06      | **27.87**            |
>
> We recommend interpreting this sweep as a stress test of cache-utilization efficiency under different KV-capacity constraints, including those that mimic consumer-grade GPUs. Even in the most cache-rich case ("No Limit"), baselines like Block-Attention saturate at ~20% hit ratio because they store *position-dependent duplicates* of the same documents. In contrast, Lazy-Attention reaches ~27.8% hit ratio (a ~40% relative improvement) thanks to its single-copy, position-agnostic design, which maximizes the effective coverage of hot documents.
>
> For deployments where the active working set still exceeds on-device KV capacity (e.g., terabytes of documents), we also implemented CPU offloading. Lazy-Attention continues to work seamlessly in this case. $\blacksquare$

---

> > ### Author Response · Authors · 2025-11-21
> >
> > **W4.** The method is tailored to RoPE-based attention. The claim that it "can easily extend to other positional encodings" is unsubstantiated. However, this is minor given that most LLMs use RoPE or RoPE-inspired PE.
> >
> > **R4.** We agree and have clarified the scope. Our method applies to positional encodings that satisfy translation invariance and group properties, i.e., only relying on the relative distance (Fact 1 in the original submission), such as standard RoPE, YaRN, NTK-scaled RoPE, and Dynamic RoPE. We have removed the broad claim about other PEs (like T5/ALiBi) and specified that Lazy-Attention is specialized for the RoPE family, which currently dominates the LLM landscape. $\blacksquare$

---

> > > ### Comment · Reviewer_Qma1 · 2025-11-27
> > >
> > > I have carefully read the rebuttal and acknowledge the authors' efforts in their rebuttal. However, I believe the answers largely confirms my initial assessment, and thus I do not find sufficient reason to adjust my original score.

---

### Official Review · Reviewer_J9c8 · 2025-11-01

**Soundness:** 2
**Presentation:** 3
**Contribution:** 2
**Rating:** 4
**Confidence:** 3

**Summary:**

Retrieval-augmented generation (RAG) faces efficiency bottlenecks due to conventional key-value (KV) caches, which embed positional encoding directly into keys/values—limiting reuse to identical document positions and requiring $O(N!)$ space for $N$ documents—while existing solutions like Block-Attention still incur KV duplication. To solve this, Lazy-Attention proposes deferred positional encoding: it stores position-agnostic KV caches (only $O(N)$ space per document) and dynamically applies relative rotation (leveraging RoPE’s relative position property) via custom Triton kernels—adding 0.59% compute overhead in prefill (rotating KV tiles once) and 0.023% average overhead in decoding (rotating queries selectively). Evaluated on NVIDIA H100 with Tulu3-Block-FT (8B) across RAG benchmarks, it reduces TTFT by up to 1.37× vs. Block-Attention (skewed distributions), boosts cache hit rates by 44% (20.69% vs. 14.38% at 10GB VRAM), maintains comparable generation quality (avg. EM 69.2%), and has only ~0.2% total overhead.

**Strengths:**

+ **Ultra-high memory efficiency and cache reuse rate**: Conventional KV caches require $O(N!)$ space for $N$ documents due to position dependence, while Lazy-Attention decouples positional encoding from KV caches via deferred encoding, making KV caches position-agnostic—each document only needs one copy of KV cache (occupying $O(N)$ space). In scenarios with 10GB VRAM and high document skewness, its cache hit rate reaches 20.69%, 44% higher than Block-Attention (14.38%), significantly improving the reuse efficiency of hot documents in RAG scenarios.
+ **Low overhead and easy engineering deployment**: It optimizes the prefill and decoding phases with custom Triton kernels—only 0.59% compute overhead in prefill and 0.023% average overhead in decoding, with a total additional overhead of ~0.2%. No model training is required; it is implemented based on PyTorch/CUDA (5,000 lines of code) and can be seamlessly integrated into existing LLM serving frameworks like vLLM, compatible with optimizations such as continuous batching, resulting in low engineering deployment costs.

**Weaknesses:**

+ **Limited experimental scope in model scale and task types**: The evaluation only uses the mid-sized Tulu3-Block-FT model (8B parameters) and focuses on RAG-oriented QA tasks (e.g., 2WikiMQA, HotpotQA), lacking validation on ultra-large LLMs (70B+ parameters) where KV cache dynamics and memory constraints may differ. It also fails to test non-technical scenarios like dialogue generation or creative writing, making it hard to confirm versatility across diverse LLM applications.
+ **Reliance on manual hyperparameter tuning**: Core parameters (e.g., KV tile size N=64, prefill query tile size M=128, decoding rotation trigger ratio r) are set manually without an adaptive adjustment mechanism. For extreme sequence lengths (e.g., <32 tokens or >1M tokens) or varying task characteristics, fixed parameters may cause redundant overhead (e.g., unnecessary rotations for short sequences) or insufficient error control (e.g., imprecise relative offsets for ultra-long sequences), increasing practical tuning costs.
+ **Unverified performance in extremely long-sequence scenarios**: While the paper validates long contexts (e.g., 4K-token documents), it does not test ultra-long sequences (e.g., millions of tokens). As sequence length scales to extreme sizes, cumulative overhead from deferred rotation may rise (e.g., more KV tiles requiring rotation), and the accuracy of relative offset calculations could degrade due to dispersed context information—issues not analyzed in the evaluation .

**Questions:**

I would be happy to increase my rating if my views are given a thorough discussion.

---

> ### Author Response · Authors · 2025-11-21
> **Thanks for your review! Here, we respond to your comments and address the issues. We hope to hear back from you if you have further questions!**
>
> **W1.** Limited experimental scope in model scale and task types.
>
> **R1.** We agree that broader coverage is important and have substantially expanded our evaluation in model scale and task types.
>
>   1. **Ultra-large model:** Llama-3.1-70B-Instruct
>   We added experiments serving Llama-3.1-70B-Instruct on a node with four H100 GPUs, under the same RAG QA workload as in Section 4.1 (5 retrieved documents per query, precomputed document KVs in DRAM, and one request per second). As shown in **Table 1**, Lazy-Attention yields TTFT speedups of 5.2× and 1.53× over standard RAG serving and CacheBlend, respectively. (Note: we evaluated TTFT with Tulu3-Block-FT 8B on a single H100 GPU node.)  We observe that the benefits are even larger on 70B than on 8B:
>   - Each token’s KV footprint in 70B is substantially larger (more layers and wider dimensions), making the KV cache budget much tighter.
>   - Lazy-Attention’s zero-copy reuse effectively saves KV memory, enabling larger batch sizes under the same HBM budget for KV cache.
>
> **Table 1**. TTFT on RAG QA for 8B vs 70B models.
> | Model | Method | TTFT (ms) | Speedup vs. Standard | Relative Advantage (Lazy vs CacheBlend) |
> | :--- | :--- | :--- | :--- | :--- |
> |Tulu3-Block-FT 8B| Standard RAG (prefix caching) | 1196.8 ms | 1.0× | - |
> || CacheBlend | 274.8 ms | 4.4× | - |
> || Lazy-Attention | 191.7 ms | 6.2× | 1.43× |
> | | | | |
> | Llama-3.1-70B (TP=4) |Standard RAG (prefix caching) | 1253.0 ms | 1.0× | - |
> ||CacheBlend | 365.5 ms | 3.4× | - |
> ||Lazy-Attention | 238.4 ms | 5.2× | **1.53×** |
>
> 2. **Non-technical / creative scenario:** Long-form literature review
>   To evaluate versatility beyond factoid QA, we added a Long-form Literature Review task:
>     - Input: 5 ArXiv papers (converted to text, 8K tokens per document on average, precomputed KV and stored in DRAM);
>     - Task: the model generates a comprehensive literature review (1024 tokens);
>     - Metric: end-to-end latency from request submission to full answer completion.
>
>     Lazy-Attention gives a 1.7× speedup, demonstrating that the method is not limited to QA and remains effective for long-context, generative tasks such as literature review / creative summarization.
> **Table 2**. Long-form literature review latency (Tulu3-Block-FT 8B, H100).
> | Method | End-to-end Latency (s) | Speedup |
> | :-- | -: | -: |
> | Standard RAG (prefix caching) | 38.0| 1.0× |
> | Lazy-Attention | 22.4  | 1.7×|
> $\blacksquare$
>
> **W2.** Reliance on manual hyperparameter tuning
>
> **R2.** We clarify that these quantities are architectural constants or algorithmically derived, not user-tuned hyperparameters, and hence impose zero manual tuning cost in practice.
>
> 1. **Tiling sizes $(N=64, M=128)$ are architectural constants.**
>    They are chosen once per GPU architecture (e.g., to match CUDA warp size and shared memory bank patterns), similarly to FlashAttention kernels, and are not adjusted per-model or per-dataset. To confirm robustness, we conducted a sensitivity study over different prefill tile sizes $M \in \{64,128,256\}$ (with $N=64$ fixed). **Table 3** reports normalized throughput.
>
> **Table 3.** Normalized throughput vs. prefill tile size $M$ (fixed $N=64$).
>    | Doc Length | M = 64 | M = 128 (Ours) | M = 256 |
>    | :- | -: |-: | -: |
>    | Short (256 tokens)  | 1.00× | 0.99× | 0.97× |
>    | Medium (4K tokens)  | 0.99× | 1.00× | 0.99× |
>    | Long   (16K tokens) | 0.98× | 1.00× | 1.00× |
>
> Throughput varies ≤3% across these choices and does *not* change any qualitative conclusion, indicating that a single fixed tiling configuration is sufficient.
>
> 2. **Rotation ratio $r$ is adaptive, not manually tuned.**
>    In decoding, we toggle on-the-fly rotation only when precomputed KV blocks are reused. The ratio
> $r \approx \frac{num_{docs} \times B}{l}$ is *computed directly from request metadata* (number of reused documents $num_{docs}$, the block size for vLLM/PagedAttention $B$, and total sequence length $l$). Users never set $r$. Therefore, long sequences (e.g., > 1M tokens), especially long documents, would be beneficial to LazyAttention, which is also the core targeted scenario for RAG. For very short sequences (e.g., <32 tokens), recomputation dominates regardless of the specific choice of $r$, which is also outside the primary target regime of RAG workloads.
>
> 3. **Numerical precision is independent of sequence length.**
>    Rather than applying incremental updates, we always rotate relative to the initial state. Concretely, the RoPE angle is computed as $\theta = (\text{logical position}) \times \omega,$
>    where $\omega$ is the base frequency. This direct-addressing scheme ensures that numerical error does not grow with sequence length; stability is independent of context length.
>
> Taken together, these design choices mean that Lazy-Attention introduces no extra hyperparameter burden for practitioners: implementation constants are fixed per hardware, and runtime behavior is determined automatically from request metadata.
> $\blacksquare$

---

> ### Author Response · Authors · 2025-11-21
>
> **W3.** Unverified performance in extremely long-sequence scenarios
>
>
> **R3.** We address both accuracy and scalability.
>
> **(1)** Accuracy of relative offsets
>
> Lazy-Attention computes RoPE positions using *exact integer arithmetic* on logical token indices. The relative offset between a precomputed KV block and its new position is derived directly from their absolute indices, making the attention mathematically equivalent to standard full attention with RoPE.
>
> To verify this empirically, we compared Lazy-Attention against standard prefix caching on long-context runs up to the model’s context limit (128K tokens) and measured differences in attention logits for the first layer (which is not affected by cross-document attention). Results are shown in **Table 4**.
>
> **Table 4.** Consistency with standard attention (H100).
>
> | Seq Length | Max abs diff in attention logits (first layer) | Max relative diff in attention logits (first layer) |
> | :--------- | -------------------------------: |-------------------------------: |
> | 128K       | 3.7457e-5  | 5.4829e-7 |
>
> These differences at 128K tokens are significantly lower than common kernel error tolerances (e.g., atol ($10^{-3}$), rtol ($10^{-5}$) in the unit test of vLLM), confirming that our position shifting is numerically stable in practice.
>
> **(2)** Scalability beyond 128K (towards 1M+ tokens)
>
> Although current public models limit us to 128K, the *per-token cost* of Lazy-Attention is independent of total sequence length:
>
> - **Stateless on-the-fly rotation.** Rotation is applied statelessly during attention reduction; the incremental cost per reused token is $O(1)$ and does not depend on the history length.
> - **Diminishing overhead ratio.** As analyzed in Section 3.2, the relative overhead is proportional to the rotation trigger ratio $r \approx \frac{num_{docs} \times B}{l}$. As the total sequence length $l$ grows (e.g., approaching millions of tokens), the denominator increases while the numerator stays bounded by the number of reused document blocks. Thus, $r$ decreases, and the relative overhead of Lazy-Attention becomes negligible for ultra-long contexts.
>
> In summary, our design provides *exact positional correctness* (matching standard RoPE attention) and *vanishing overhead* as sequence length grows. We have empirically validated correctness and efficiency up to 128K tokens and analytically argued scalability beyond that range. $\blacksquare$
>
> **Q1.** "I would be happy to increase my rating if my views are given a thorough discussion."
>
> **A1.** We sincerely appreciate the reviewer’s openness to reconsidering the rating. We hope that the additional experiments and clarifications above address the core concerns:
>
> - **Expanded scope (W1):** We added results on **Llama-3.1-70B-Instruct** and a **long-form literature review** task, showing consistent TTFT improvements and latency reduction beyond QA.
> - **No manual tuning (W2):** We clarified that tiling parameters are *architectural constants*, and the rotation ratio $r$ is *derived automatically* from request metadata, with no user-tuned hyperparameters.
> - **Scalability and correctness (W3):** We demonstrated numerically stable outputs up to 128K tokens and provided a complexity analysis showing that the relative overhead of Lazy-Attention decreases as sequence length grows.
>
> We would be grateful if you could reconsider your rating in light of this additional evidence. $\blacksquare$

---

> > ### Comment · Reviewer_J9c8 · 2025-11-26
> >
> > Thanks for your replay, I have increased the score.

---

> > > ### Author Response · Authors · 2025-11-26
> > >
> > > We sincerely appreciate your review and your positive feedback on our work!

---

### Official Review · Reviewer_4N3e · 2025-11-02

**Soundness:** 3
**Presentation:** 3
**Contribution:** 2
**Rating:** 4
**Confidence:** 3

**Summary:**

The paper proposes a lazy or deferred attention scheme: store position-free K/V in the cache and apply RoPE (or more generally, the position transform) inside the attention kernel at lookup time, using the query–key relative offset. This lets different requests reuse the same per-document KV even when the document appears at different positions. The paper further gives two Triton kernels (prefill vs. decode) and integrates it into a vLLM-style runtime to show better TTFT and higher KV hit ratio under skewed RAG workloads.

**Strengths:**

- Addresses a very real, very current bottleneck in RAG-serving: repeated passages at different offsets.

- Moves positional handling into the kernel in a way that is compatible with existing vLLM runtimes and shows low overhead on real hardware.

- Experimental evidence broadly matches the theoretical overhead story (small prefill cost, tiny decode cost) and shows higher KV hit ratio under skewed RAG.

- Engineering is neat: two kernels, offset packing, and a clear explanation of where the extra FLOPs go.

**Weaknesses:**

- Novelty is narrower than claimed. There are already RAG-oriented KV reusers (RAGCache) and KV-centric serving systems (Mooncake). There are already attention mechanisms that explicitly decouple RoPE from KV (MLA / TransMLA / DeepSeek-V2/V3). There is already block-wise RAG reuse that solves the “position doesn’t match” problem, albeit by re-encoding and fine-tuning (Block-Attn). Your paper’s real addition is to push that decoupling into a production kernel and make it work without retraining. That’s incremental.

- Baselines omit some of the most relevant contemporary systems, so the empirical advantage is not yet airtight.

- Scope is a bit tailored: H100 + vLLM + RAG with hot docs. It’s not yet shown this stays cheap on weaker GPUs or less skewed workloads.

- Related work section needs to be rewritten to explicitly place the method next to RAGCache (tree-structured reuse), Mooncake (disaggregated KV), vLLM PagedAttention (logical–physical decoupling), and MLA/TransMLA (decoupled RoPE). Right now it overstates novelty.

**Questions:**

Your idea becomes much more convincing when framed as "kernelizing the known RoPE-decoupling trick for production RAG engines" rather than "we are the first to let multiple prompts share KV at different positions". The former is true; the latter is not.

---

> ### Author Response · Authors · 2025-11-21
> **Thanks for your review! Here, we respond to your comments and address the issues. We hope to hear back from you if you have further questions!**
>
> **W1.** Novelty is narrower than claimed. The paper’s real addition is to push that decoupling into a production kernel and make it work without retraining incrementally.
>
> **R1:** We thank the reviewer for the detailed contextualization. We agree that the high-level concepts of KV reuse and RoPE decoupling exist in prior literature. However, while we agree that the high-level ideas of KV reuse and RoPE decoupling are not new, we believe our work goes beyond an incremental implementation: the core contribution of Lazy-Attention is to resolve the *Memory–Compute trade-off* that has so far prevented these ideas from scaling to high-throughput serving.
>
> Prior works generally fall into three categories: (i) restrict reuse to prefixes, causing memory fragmentation; (ii) allow arbitrary reuse but require *memory materialization (copying)*; or (iii) require specific model architectures (MLA).
> To the best of our knowledge, Lazy-Attention is the first system to devise RoPE decoupling as a zero-copy, arbitrary-position KV reuse kernel on standard architectures, resolving the memory–compute trade-off that prior methods leave open in practice.
>
> We provide a detailed comparison in **Table 1**.
>
> **Table 1.** Comparison of KV-Reuse and RoPE-Decoupling methods.
>
> | Method | Reuse Flexibility | Memory Overhead | Compute Cost | Model Requirement |
> | :--- | :--- | :--- | :--- | :--- |
> | RAGCache$^1$ / Mooncake$^2$ | Prefix Only | High (Duplicate Copy) | High (Re-computation) | Standard Arch |
> | CacheBlend$^3$ / EPIC$^4$ | Any Position | High (Duplicate Copy) | Mid (Partial Re-compute) | Standard Arch |
> | TurboRAG$^5$ / Block-Attn$^6$  | Any Position | High (Duplicate Copy) | Low (Re-encoding) | Standard Arch |
> | MLA$^{7,8}$ / TransMLA$^{9}$ | Prefix Only | Low (Compressed) | Low (Projection) | Specific Arch Only |
> | Lazy-Attention (Ours) | Any Position | Zero (Shared Copy) | Negligible (\<1% in our exps) | Standard Arch |
>
>
> **1. Distinction from RAG-oriented Caching (RAGCache, Mooncake)**
> These systems rely on *Prefix Caching*. If a document appears in a different position (e.g., shifted by a system prompt), these systems cannot reuse the cached KV and must re-generate the states from scratch and store a *duplicate copy* in HBM. Lazy-Attention decouples position from storage, allowing a single physical copy to serve logically distinct requests.
>
> **2. Distinction from Re-encoding Methods (Block-Attn, TurboRAG)**
> This is a critical distinction. While Block-Attn and TurboRAG achieve position flexibility with *low computational cost* for re-encoding, they are fundamentally limited by *memory materialization*. They must write the re-encoded KV states into a new memory buffer to perform attention. This consumes HBM capacity and memory bandwidth. In contrast, Lazy-Attention fuses the re-encoding into the attention kernel *on-the-fly*. This eliminates the need for duplicate memory copies and avoids the bandwidth overhead of writing back to HBM.
>
> **3. Distinction from MLA / TransMLA**
> MLA utilizes a "Decoupled RoPE" strategy, but its primary objective is *KV compression* (via low-rank projection), not position-agnostic reuse. In MLA, RoPE is applied to a subset of dimensions to preserve the low-rank structure of the content part, but the attention computation still relies on absolute positional alignment. Consequently, MLA-based systems remain restricted to *Prefix Caching* paradigms (similar to RAGCache) and cannot reuse KV blocks when their positions change within the prompt. In contrast, Lazy-Attention enables zero-copy sharing of KV blocks at arbitrary positions.
>
> We revised the Related Work section to explicitly clarify these differences, emphasizing that our production kernel is the basis for *Zero-Copy* reuse.
>
> Reference:
>
> 1. Jin, Chao, et al. "Ragcache: Efficient knowledge caching for retrieval-augmented generation." ACM Transactions on Computer Systems (2024).
> 2. Qin, Ruoyu, et al. "Mooncake: A kvcache-centric disaggregated architecture for llm serving." ACM Transactions on Storage (2024).
> 3. Yao, Jiayi, et al. "CacheBlend: Fast large language model serving for RAG with cached knowledge fusion." EuroSys. 2025.
> 4. Hu, Junhao, et al. "EPIC: Efficient Position-Independent Caching for Serving Large Language Models." ICML. 2025.
> 5. Lu, Songshuo, et al. "Turborag: Accelerating retrieval-augmented generation with precomputed kv caches for chunked text." EMNLP. 2025.
> 6. Ma, Dongyang, Yan Wang, and Tian Lan. "Block-Attention for Efficient Prefilling." ICLR. 2025.
> 7. Liu, Aixin, et al. "Deepseek-v2: A strong, economical, and efficient mixture-of-experts language model." arXiv preprint arXiv:2405.04434 (2024).
> 8. Liu, Aixin, et al. "Deepseek-v3 technical report." arXiv preprint arXiv:2412.19437 (2024).
> 9. Meng, Fanxu, et al. "Transmla: Multi-head latent attention is all you need." arXiv preprint arXiv:2502.07864 (2025).
>
> $\blacksquare$

---

> ### Author Response · Authors · 2025-11-21
>
> **W2**. Baselines omit some of the most relevant contemporary systems.
>
> **R2:** We appreciate the reviewer’s suggestion regarding additional baselines. **(i) RAGCache Implementation:** As requested, we implemented a baseline representing RAGCache, specifically incorporating its core *Knowledge Tree-based prefix-aware Greedy-Dual-Size-Frequency (PGDSF) eviction policy*. We note that RAGCache's contribution is primarily orthogonal to ours: it optimizes cache management (what to keep), whereas Lazy-Attention optimizes cache utilization (where it can be used). **(ii) Performance Comparison:** **Table 2** (please refer to the second and fourth rows) demonstrates performance under the same setting of Section 4.1 with uniform sampling (Each request references 5 documents whose per-document KV caches have been precomputed and stored in DRAM, and the request rate is one request per second). Even with the advanced PGDSF policy, RAGCache fails to reuse cached blocks when positions shift. But Lazy-Attention effectively identifies reuse opportunities regardless of position, achieving a $5.76 \times$ lower TTFT compared to RAGCache (191.7 ms vs. 1104.5 ms). **(iii) Mooncake:** Similarly, Mooncake relies on prefix-caching (like RAGCache) within a prefill-decoding disaggregated architecture. Therefore, the limitations observed in the RAGCache baseline apply equally to Mooncake.
>
> **Table 2.** Comparison under Uniform Sampled Documents.
> | Method | Cache Reuse Logic | TTFT (ms) |
> | :--- | :--- | :--- |
> | Prefix Caching | Prefix Match | 1196.8ms |
> | RAGCache (Prefix Caching with PGDSF) | Prefix Match | 1104.5ms |
> | CacheBlend | Position Agnostic | 274.8ms|
> | Lazy-Attention | Position Agnostic | 191.7 ms |
>
> $\blacksquare$
>
>
> **W3.** Scope is a bit tailored for "H100 + vLLM + RAG with hot docs".
>
> **R3:** We appreciate the reviewer’s comment regarding generalizability. We address this in two parts, explicitly considering our system setting where per-document KV caches are precomputed and resident in DRAM.
>
> **(1) Generalization to Weaker GPUs:**
> We acknowledge that H100 is high-end. However, in a system serving precomputed KVs, the bottleneck is primarily Memory Bandwidth, not Compute.
> * **Why weaker GPUs suffer more:** Lower-tier GPUs (e.g., A40/A100) have significantly lower bandwidth than H100. Existing methods like Block-Attn require *materializing* the adapted KVs—reading the precomputed block from DRAM and *writing a new copy back to DRAM* to ensure contiguity or alignment. This "Read-Modify-Write" pattern consumes double the memory bandwidth.
> * **Lazy-Attention Advantage:** Our method performs RoPE fusion on-the-fly, keeping KV access strictly read-only with zero additional KV writes (beyond the final attention outputs).
> * **Results:** Our new experiments on *NVIDIA A40 (48GB) and A100 (40GB)* confirm this (please refer to **Table 3**). In fact, the relative speedup of Lazy-Attention over other position-agnostic methods like CacheBlend is strengthened on bandwidth-constrained GPUs, where saving the extra write overhead becomes critical once memory bandwidth is the bottleneck. This trend is summarized in the last column of Table 3: the relative gain of Lazy-Attention over CacheBlend increases from 1.43× on H100 to 1.51× on A100 and 1.70× on the bandwidth-limited A40.
>
> **Table 3.** TTFT Comparison under Uniform Sampled Documents across different GPUs.
>
> | Hardware | Method | TTFT (ms) | Speedup (vs Prefix) | Relative Speedup (Lazy vs CacheBlend) |
> | :--- | :--- | :--- | :--- | :--- |
> | **H100** | Prefix Caching | 1196.8 ms | 1.0x | - |
> | *(High Bandwidth)* | RAGCache (PGDSF) | 1104.5 ms | 1.08x | - |
> | | CacheBlend | 274.8 ms | 4.3x | - |
> | | **Lazy-Attention** | **191.7 ms** | **6.2x** | **1.43x** |
> | | | | | |
> | **A100** | Prefix Caching | 2580.4 ms | 1.0x | - |
> | *(Mid Bandwidth)* | RAGCache (PGDSF) | 2390.1 ms | 1.08x | - |
> | | CacheBlend | 565.3 ms | 4.5x | - |
> | | **Lazy-Attention** | **372.5 ms** | **6.9x** | **1.51x** |
> | | | | | |
> | **A40** | Prefix Caching | 4150.5 ms | 1.0x | - |
> | *(Low Bandwidth)* | RAGCache (PGDSF) | 3820.2 ms | 1.08x | - |
> | | CacheBlend | 1150.6 ms | 3.6x | - |
> | | **Lazy-Attention** | **675.2 ms** | **6.1x** | **1.70x** |
>
> **(2) Robustness to Less Skewed Workloads:** Figure 3(a) evaluates a strictly Uniform Distribution (Zero Skew). In this worst-case scenario, reuse is rare. However, Lazy-Attention stays cheap and robust while baselines degrade. As shown in Figure 3(a) (rightmost points), as request rates increase, Block-Attention suffers from sharp latency spikes. This is due to the "Read-Modify-Write" overhead of materialization from repeatedly writing re-encoded blocks back. In contrast, Lazy-Attention remains strictly Read-Only regarding KV access, avoiding bandwidth saturation and maintaining consistent latency even at high throughput.
> $\blacksquare$

---

> ### Author Response · Authors · 2025-11-21
>
> **W4.** Related work section needs to be rewritten.
>
> **R4:** We thank the reviewer for the insightful categorization. We rewritten the Related Work section to strictly contextualize Lazy-Attention within the existing ecosystem. The revised section focuses on precise differentiation rather than broad claims:
>
> 1.  Vs. vLLM/PagedAttention: We frame our method as extending the decoupling paradigm introduced by vLLM. While PagedAttention decouples logical/physical addresses to manage memory fragmentation, we apply this decoupling concept to positional semantics to enable reuse.
> 2.  Vs. RAGCache/Mooncake: We clarify that our work is *complementary* to these systems. They focus on *Cache Management* (eviction policies like PGDSF), while we focus on the underlying *Cache Utilization* mechanism (zero-copy reuse).
> 3.  Vs. MLA: We distinguish MLA as a model-level architectural design for compression, whereas our approach is a system-level optimization for standard architectures.
>
> This restructuring ensures a fair comparison, highlighting that Lazy-Attention is a specific solution for *zero-copy reuse* that fits into the broader landscape defined by these works.
> $\blacksquare$
>
> **Q1:** The idea becomes much more convincing when framed as "kernelizing the known RoPE-decoupling trick for production RAG engines".
>
> **A1:** We agree and adopted this precise framing in the revision. We now characterize our contribution as kernelizing the RoPE-decoupling technique to enable efficient, zero-copy reuse in production systems, rather than claiming novelty for the conceptual decoupling itself. By pushing this logic into a fused kernel, we resolve the materialization overhead that previously hindered the practical deployment of this technique. We carefully revised our abstract and Introduction section to clarify the kernerlizing efforts and the goal of zero-copy reuse. $\blacksquare$

---

### Comment · Area_Chair_hD9S · 2025-11-28

Dear Reviewers,

The authors have responded to your reviews. Please engage in the discussion and evaluate the authors’ rebuttal to check whether your comments have been adequately addressed, and determine whether you would like to adjust your evaluations.

Best,

Your AC

---

### Author Response · Authors · 2025-11-28
**Rebuttal Summary**

We thank the reviewers for their constructive engagement. We are encouraged that Reviewer `J9c8` raised their score to 6 following our new experiments on scalability, and Reviewer `Qma1` acknowledged the "very valid engineering contribution" of this work.

**1. New Experiments: Verified Scalability & Generalization**
Motivated by the valuable comments, we added comprehensive results in Appendix F to demonstrate that Lazy-Attention meets modern production requirements:
- **Scalability to 70B Models:** Deployed on Llama-3.1-70B (4×H100), achieving a 5.2× speedup in TTFT (1.53× vs. CacheBlend). This proves our Zero-Copy benefit amplifies on large models where memory bandwidth is the primary bottleneck.
- **Universal Compatibility:** Verified effectiveness on Qwen3 (standard RoPE) and Lego-Link0 (training-free reuse), confirming our kernels work across diverse architectures and reuse strategies without retraining.
- **Long-Context Stability:** Verified accuracy up to 128K tokens and performance on 16K-token documents. Attention logits match reference attention ($<10^{-5}$ diff), confirming no cumulative RoPE error.
- **Hardware Robustness:** Demonstrated consistent speedups on bandwidth-constrained NVIDIA A40 and A100 GPUs, significantly outperforming baselines like RAGCache (191ms vs 1104ms TTFT).

**2. Core Contribution: From Algorithmic Concept to System Reality**
We respectfully clarify the distinct system-level contribution of Lazy-Attention compared to pioneering re-encoding approaches (e.g., TurboRAG, Block-Attention):
- **Solving the Materialization Bottleneck:** Prior works demonstrate _algorithmic validity_ but often require "Read-Modify-Write" materialization. This copies states to new buffers, consuming critical HBM bandwidth and causing cache fragmentation before decoding even begins.
- **The Zero-Copy Solution:** Lazy-Attention fuses position adjustment on-the-fly. We enforce strict physical immutability, allowing a single shared KV block to serve infinite logical positions without fragmentation.
- **System-Algorithm Co-design:** Recognizing that decoding is constrained by the Memory Wall rather than FLOPs, we shift the bottleneck from memory access to lightweight on-chip computation (_rotating Q instead of mutating K_). This ensures algorithmic flexibility scales efficiently on modern accelerators.

Lazy-Attention bridges the gap between algorithmic innovations and production-grade serving. We believe this system-kernel co-design offers valuable insights for the Efficient ML and SysML community by ensuring flexibility does not come at the cost of hardware efficiency.

---

### Author Response · Authors · 2025-12-02
**Author Final Remarks**

To assist the new AC following the review restructuring, we highlight our core system contribution and the resulting consensus among reviewers.

**Contribution.** Reducing TTFT through KV caching is an active research direction. Existing approaches (e.g., [BlockAttn](https://openreview.net/forum?id=7zNYY1E2fq), [TurboRAG](https://aclanthology.org/2025.emnlp-main.334/), [CacheBlend](https://dl.acm.org/doi/10.1145/3689031.3696098)) introduce mathematical operations for positional re-encoding, but none consider efficient algorithm and hardware implementation, leading to unnecessary data movement and materialization overheads.
To the best of our knowledge, we are the first to algorithmically defer the RoPE rotation based on relative distance and kernelize positional adjustment by fusing it into the attention kernel. This design keeps the KV cache entirely position-agnostic (Zero-Copy), resolving the underlying memory–compute trade-off. Thus, we achieved up to 14.4x VRAM hit ratio vs Standard Prefix Caching (1.4x VRAM hit ratio vs the state-of-the-art baseline) with negligible overhead.

**Review.** Reviewers recognized the value of this system-algorithm co-design. Reviewers **Qma1** and **4N3e** acknowledged the work as a "very valid engineering contribution" that addresses a "very real bottleneck." Notably, Reviewer **J9c8** explicitly acknowledged our rebuttal with the results in scalability and raised their score to 6.

Reviewers also pointed out potential weaknesses regarding model scale, hardware diversity, and context length. Within the rebuttal period, we addressed all of them. Specifically:
- Scalability (Reviewer **J9c8**): We expanded evaluation to Llama-3.1-70B (4xH100). We achieved a 1.53x TTFT speedup vs CacheBlend (5.2x vs Standard), proving that our Zero-Copy benefit amplifies on large models where memory bandwidth is the primary bottleneck.
- Hardware Robustness (Reviewer **4N3e**): We added experiments on NVIDIA A40/A100. On the bandwidth-constrained A40, our relative speedup over CacheBlend increased to 1.70x (1.51x on A100 vs 1.43x on H100), empirically confirming the architectural advantage of avoiding materialization.
- Long Context (Reviewer **Qrx9**): We validated performance on 16K-token documents and verified numerical stability up to 128K tokens (max logit diff $< 10^{-5}$), ensuring no cumulative errors.

With these additional validations across diverse hardware and large models, we believe our approach demonstrates its value as a robust, production-ready solution. We are grateful for the feedback that helped solidify the paper for the final version.

---

### Meta-Review · Area_Chair_q3af · 2026-01-06

**Summary:**

This paper proposes a mechanism that defers positional encoding in Transformers to enable position-agnostic KV cache reuse in RAG systems. The key idea is to apply RoPE rotations during attention computation rather than before caching, allowing document caches to be shared across different prompt positions without duplication. The reviewers propose the following key concerns:
1. The novelty is narrower than claimed, particularly, the core insight of deferring RoPE application is from Block-Attention.
2. The experiments are limited:
 1) the performance in (extremely) long-sequence scenarios are not reported
 2) the evaluation uses only one model, which is specifically fine-tuned for Block-Attention. Generalization to other architectures unclear.
3. Lazy-Attention excels with hot documents but struggles with cold starts (no cached documents). In uniform distributions (low reuse), its prefill (0.59%) and decoding (0.023%) overheads make it only slightly better than Block Attention (TTFT difference <5%), with no advantage over Prefix Caching for fully cold request

**Reviewer Concerns:**

1. After rebuttal, the experiments concern is largely addressed as the authors show further experiments on different models such as Tulu3-BLock-FT, Qwen3-8B-Instruct, Lego-link0, and extend the sequence from 4k to 16k.
2. The novelty problem is still outstanding, as the authors also agree the mathematical intuition of deferring RoPE shares roots with Block-Attention, and further adjust their contribution as system-kernel co-design (but this indeed is largely due to the engineering techniques such as kernel fusion). I appreciate the engineering contribution of this work, but this also confirms the incremental novelty.
3. While the VRAM cache hit ratio shows substantial improvement compared with Block Attention, the TTFT difference shows no big difference, and I did not find the corresponding answer in the rebuttal to this point.

**Reviewer Scores:**

The original scores are 4(4N3e), 4(J9c8), 6(Qma1), and 4(Qrx9). After read the rebuttals and the response from the reviewers (only part of them due to the known reason), I would expect the scores to change to 4(4N3e), 6(J9c8), 6(Qma1), and 4(Qrx9).

---

### Decision · Program_Chairs · 2026-01-26

Reject